# Intrinsically disordered CsoS2 acts as a general molecular thread for α-carboxysome shell assembly

Tao Ni [1,2,7] ✉, Qiuyao Jiang[3,7], Pei Cing Ng [3], Juan Shen[1], Hao Dou[1], Yanan Zhu [1], Julika Radecke [4], Gregory F. Dykes [3], Fang Huang[3], Lu-Ning Liu [3,5] ✉ & Peijun Zhang [1,4,6] ✉

Carboxysomes are a paradigm of self-assembling proteinaceous organelles found in nature, offering compartmentalisation of enzymes and pathways to enhance carbon fixation. In α-carboxysomes, the disordered linker protein CsoS2 plays an essential role in carboxysome assembly and Rubisco encapsulation. Its mechanism of action, however, is not fully understood. Here we synthetically engineer α-carboxysome shells using minimal shell components and determine cryoEM structures of these to decipher the principle of shell assembly and encapsulation. The structures reveal that the intrinsically disordered CsoS2 C-terminus is well-structured and acts as a universal "molecular thread" stitching through multiple shell protein interfaces. We further uncover in CsoS2 a highly conserved repetitive key interaction motif, [IV]TG, which is critical to the shell assembly and architecture. Our study provides a general mechanism for the CsoS2-governed carboxysome shell assembly and cargo encapsulation and further advances synthetic engineering of carboxysomes for diverse biotechnological applications.

Organelles confine specific biochemical pathways within the cell to enhance metabolic efficiency, alleviate metabolic crosstalk, and facilitate spatiotemporal regulation of sequestered pathways[1]. Apart from eukaryotes, in the past decades, advances in bioinformatics, biochemistry, imaging, and cell physiology have demonstrated that bacteria have also evolved subcellular organelles, including bacterial microcompartments (BMCs) which is composed entirely of proteins, to compartmentalize metabolism[2].

Carboxysomes are anabolic BMCs for autotrophic $CO_2$ fixation found in all cyanobacteria and many proteobacteria[3–7]. The carboxysome is composed of a polyhedral shell that encapsulates the key $CO_2$-fixation enzyme, ribulose-1,5-bisphosphate carboxylase/oxygenase

(Rubisco), and carbonic anhydrase (CA), which dehydrates $HCO_3^-$ to $CO_2$, the substrate for Rubisco carboxylation[6,8–18]. The carboxysome shell acts as a selectively permeable barrier, allowing the influx of $HCO_3^-$ and ribulose 1,5-bisphosphate (RuBP) while presumably precluding $O_2$ influx and $CO_2$ leakage[19–23]. The intriguing structural features of carboxysomes are fundamental for maximizing $CO_2$ assimilation and reducing the unproductive Rubisco oxygenation, thereby allowing carboxysomes to make substantial contributions to the global carbon fixation and primary production[13].

The α-carboxysome shell comprises predominantly CsoS1 hexamers that form the shell facets and CsoS4 pentamers that occupy the vertices of the polyhedral shell, both hexamers and pentamers

[1]Division of Structural Biology, Wellcome Trust Centre for Human Genetics, University of Oxford, Oxford OX3 7BN, UK. [2]School of Biomedical Sciences, LKS Faculty of Medicine, The University of Hong Kong, Pokfulam, Hong Kong SAR, China. [3]Institute of Systems, Molecular and Integrative Biology, University of Liverpool, Liverpool L69 7ZB, UK. [4]Diamond Light Source, Harwell Science and Innovation Campus, Didcot OX11 0DE, UK. [5]College of Marine Life Sciences, and Frontiers Science Center for Deep Ocean Multispheres and Earth System, Ocean University of China, Qingdao 266003, China. [6]Chinese Academy of Medical Sciences Oxford Institute, University of Oxford, Oxford OX3 7BN, UK. [7]These authors contributed equally: Tao Ni, Qiuyao Jiang.
✉e-mail: taoni@hku.hk; Luning.Liu@liverpool.ac.uk; peijun.zhang@strubi.ox.ac.uk

containing multiple paralogous proteins[24–26]. An intrinsically disordered protein CsoS2 in high abundance[24] serves as a linker protein to bind both Rubisco and the shell through its N-terminal (CsoS2-N) and C-terminal domains (CsoS2-C), respectively[11,27–30]. It is presumed that α-carboxysome biogenesis adopts the 'Partial shell first' or 'Concomitant shell–core' assembly pathways[6,31,32], and CsoS2 is essential for α-carboxysome biogenesis and assembly of intact α-carboxysome shell[33]. However, how CsoS2 interacts with the shell and governs shell assembly remains enigmatic. Synthetic BMC shells provide a means for investigating the assembly mechanisms and pairwise interactions that drive shell formation[33–39]. They also hold promises for generating new caging nanomaterials in new contexts, such as enzyme encapsulation, molecule scaffolding and delivery[40,41].

In this work, we synthetically engineer α-carboxysome shells using minimal shell components derived from a chemoautotroph *Halothiobacillus* (*H.*) *neapolitanus*, a model system in the fundamental studies and synthetic engineering[11,21,24,42–49], and determine high-resolution cryoEM structures of the shells of variable constituents. Surprisingly, the structures show that the intrinsically disordered CsoS2-C makes well-defined multivalent contacts with shell proteins, functioning as a "molecular thread" to stitch the assembly interfaces and mediate shell assembly. Moreover, we uncover a remarkable key repeating motif critical to the assembly and architecture of the shell. These findings provide insight into the CsoS2-mediated assembly principles of α-carboxysome shell.

## Results and discussion
### Assembly and structures of recombinant α-carboxysome shells
Expressing all shell components encoded by genes in the *cso* operon (Fig. 1a, coloured red, purple, and blue) results in formation of native-like α-carboxysome shells[33]. To investigate the molecular principles of α-carboxysome shell formation and the role of CsoS2 in α-carboxysome shell assembly, we took a reductionist approach and designed two shell constructs using minimal shell components, CsoS1A (a CsoS1 paralog) and CsoS4A (a CsoS4 paralog) for mini-shell 1, and CsoS1A and CsoS4A with additional CsoS2 (encoded by the natural *Halothiobacillus csoS2* gene that contain a ribosomal frame-shifting region[28,50]) for mini-shell 2 (Fig. 1b, Supplementary Fig. 1a). Expression of either construct in *Escherichia coli* (*E. coli*) produces assembled shell architectures (Supplementary Fig. 2a, c), while expressing of mini-shell 3 (CsoS1A and CsoS2 without CsoS4A) does not yield any assembly (Supplementary Fig. 1a, b). These results indicate that CsoS1A and CsoS4A together are sufficient to form shell assemblies. However, dynamic light scattering (DLS) reveals that, while mini-shell 1 produces shells of ~23 nm in diameter, mini-shell 2 generates predominantly larger shells of ~35 nm in diameter, in addition to the shells of ~23 nm in diameter (Supplementary Fig. 2b). CryoEM analysis further establishes an icosahedral architecture for the shell assemblies and reveals that mini-shell 1 assemblies contain mainly small shells (~21 nm, $T = 3$; see definition of T number is described in Methods), whereas mini-shell 2 products contain mostly large shells (~35 nm, $T = 9$) (Fig. 1c, Supplementary Fig. 3a, b). A medium sized shell (~23 nm, $T = 4$) is present in both constructs in low abundance. It is intriguing that the large $T = 9$ shells are only present in CsoS2-containing mini-shell 2.

We determined the structures of these $T = 9$, $T = 4$, and $T = 3$ shell assemblies at 1.86 Å, 3.54 Å and 2.79 Å resolution, respectively (Fig. 1d, Supplementary Table 1, Supplementary Fig. 3c–f). The two $T = 4$ shell structures from both mini-shell constructs (Fig. 1c) appear largely identical, except that the β4-α2 loop of CsoS1A from mini-shell 1 is disordered (Supplementary Fig. 4a). All the shell proteins possess the same concave-out orientation, consistent with other shell assemblies[34,37,38]. Intriguingly, only $T = 9$ shells generated from mini-shell 2 contain extra densities not accounted for by CsoS1A and

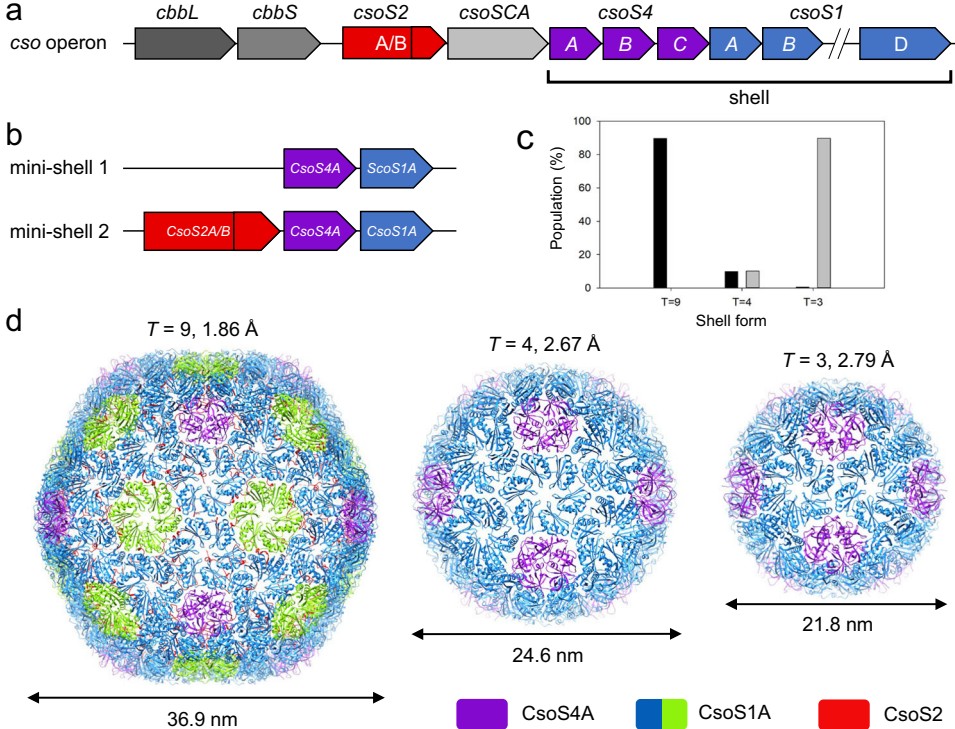

**Fig. 1 | Design and overall cryoEM structure of α-carboxysome mini-shells.** **a** Schematics of α-carboxysome *cso* operon. **b** Schematics of two mini-shell constructs used to assemble shells. **c** Distribution of shell forms assembled with mini-shell construct 1 (gray, total 177,237 mini-shells) and mini-shell construct 2 (black, total 137,690 mini-shells). Source data are provided as a Source data file. **d** CryoEM structures of three different shell forms with icosahedral symmetry of $T = 9$, $T = 4$ and $T = 3$, at the resolution of 1.86 Å, 2.67 Å, and 2.79 Å, respectively. The diameter of shells are indicated. Shell components are coloured in purple (CsoS4A pentamer), blue/green (quasi-equivalent CsoS1A hexamer) and red (CsoS2) which is only present in the $T = 9$ shell.

CsoS4A at the inner surface of the shell, which are absent in $T = 4$ and $T = 3$ shells. The high-quality map of the $T = 9$ shell (1.86 Å resolution) allows for accurate modelling of CsoS2-C that tightly associates with shell proteins, resulting in an atomic model of the $T = 9$ shell comprising 80 CsoS1A hexamers, 12 CsoS4A pentamers and 60 CsoS2-C (Fig. 1d).

## Structural plasticity of shell proteins and protein-protein interactions

Three different sized shells with $T = 3$, $T = 4$ and $T = 9$ icosahedral symmetries are built of essentially the same hexameric CsoS1A and pentameric CsoS4A (Fig. 1d), which are highly conserved across α-cyanobacteria and many proteobacteria (Supplementary Fig. 5a). The RMSDs of the basic assembly units (a.k.a. capsomeres), CsoS1A hexamer and CsoS4A pentamer, in three icosahedral symmetries range 0.180-0.231 Å and 0.240-0.251 Å (calculated from the pairwise comparison of Cα), respectively. Superimposing the cryoEM structures of these capsomeres with their X-ray crystal structures (PDB: 2EWH, PDB: 2RCF) reveals subtle deviations of curvature within hexamer (the crystal structure closely resembles the least curved $T = 9$), and largely identical pentamer (Supplementary Fig. 5b), with the overall RMSD range of 0.321–0.597 and 0.204–0.417, respectively.

There are four inter-capsomere assembly interfaces in α-carboxysome shells (Fig. 2a, interface 1–4). Of those, interfaces 3 and 4 are unique to the $T = 9$ shell, whereas interfaces 1 and 2 are present in all three shells as previously reported[38]. The angles between capsomeres in interfaces 1 and 2 vary slightly among three shells, from 30° to 35° and 30° to 43°, respectively (Fig. 2b, c), leading to small curvature changes in the shell assembly, thus the size differences. In contrast, the angles between two hexamers in interfaces 3 and 4 are arranged in a planar fashion, markedly different from those in interfaces 1 and 2 (Fig. 2d), and also different from those measured in other BMC mini-shells[37]. Despite a large deviation of the tilt angle (from 0° to 43°), the main interactions at the interface between adjacent hexamers, which is mediated by the hydrogen bond network involving Lys29,

Ala30, and Arg83, remain largely conserved (Fig. 2e). Collectively, the results suggest that the plasticity of inter-capsomere interfaces contributes to the curvature and thus, the structural polymorphism of carboxysome shells.

## CsoS2 is a molecular thread linking multiple capsomeres

CsoS2 protein is present in α-cyanobacteria, α-proteobacteria, β-proteobacteria, and γ-proteobacteria (Supplementary Fig. 6). It functions as the linker protein connecting cargo enzymes to the shell and is a vital component in α-carboxysome assembly[24,27,33]. Genetic deletion of *csoS2* resulted in loss of carboxysomes in the *H. neapolitanus* cells and high CO2-requiring phenotypes[28], and recombinant intact shells could not be formed in the absence of CsoS2[33]. CsoS2 is a large polypeptide (~900 residues) composed of three regions: a N-terminal region (CsoS2-N), a middle region (CsoS2-M), and a C-terminal region (CsoS2-C)[28,29,33] (Fig. 3a). Repetitive short linear motifs have been identified in CsoS2, which vary in numbers among species; for example, the *H. neapolitanus* CsoS2 contains 4 N-repeats, 6 M-repeats, and 3 C-repeats[5] (Fig. 3a, Supplementary Fig. 1c). Structure prediction by AlphaFold2 revealed that CsoS2 represents a largely disordered protein, especially in the C-terminal region (Supplementary Fig. 7b), consistent with previous analyses[27,28]. Recent studies have shown that the CsoS2 N-terminal domain binds with Rubisco, playing roles in mediating Rubisco encapsulation[11,27], whereas the C-terminus of CsoS2 binds with the shell and could serve as an encapsulation anchor for cargo recruitment[28,33]. However, how CsoS2 anchors to the shell and whether it plays a role in governing shell assembly have remained enigmatic.

The mini-shell 2 vector comprises the native *csoS2* gene *Halothiobacillus*, which contains a ribosomal frameshifting region thereby resulting in the production of two CsoS2 isoforms, the full-length CsoS2B and the C-terminus-truncated CsoS2A. Our immunoblot analysis revealed that both CsoS2A and CsoS2B isoforms were expressed in the *E. coli* mini-shell construct (Supplementary Fig. 2b), and the ratio of CsoS2A and CsoS2B is comparable to that found in the native α-carboxysome from *Halothiobacillus*[24]. By contrast, only CsoS2B was

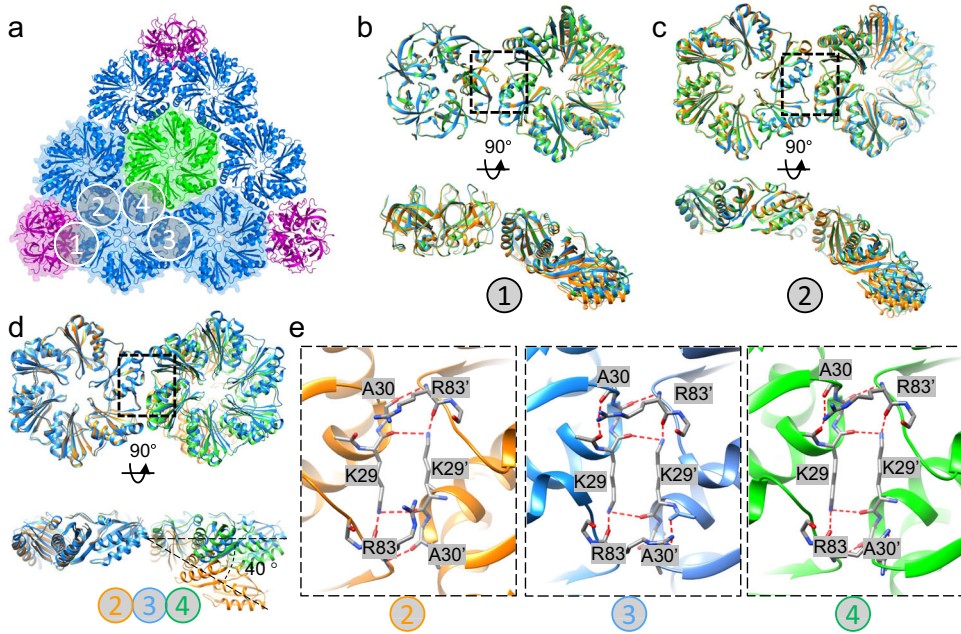

**Fig. 2 | Structurally conserved shell proteins with plastic assembly interfaces. a** The overall organization $T = 9$ shell, with assembly interfaces 1–4 between capsomeres labelled. Interfaces 3 and 4 are unique to $T = 9$. **b-c** Overlay of interface 1 (**b**) and interface 2 (**c**) from $T = 9$ (blue), $T = 4$ (green) and $T = 3$ (orange) shells, viewed from top (left) and side (right). **d** Overlay of interfaces 2 (orange), 3 (blue) and 4 (green) from $T = 9$ shell only, aligned to the shared hexamer. There is a ~40 ° difference in curvature. **e** Details of interacting residues in the dimer interfaces 2, 3 and 4 in the $T = 9$ shell (dashed box in (**d**)). The hydrogen bond network is labelled with red dashed line.

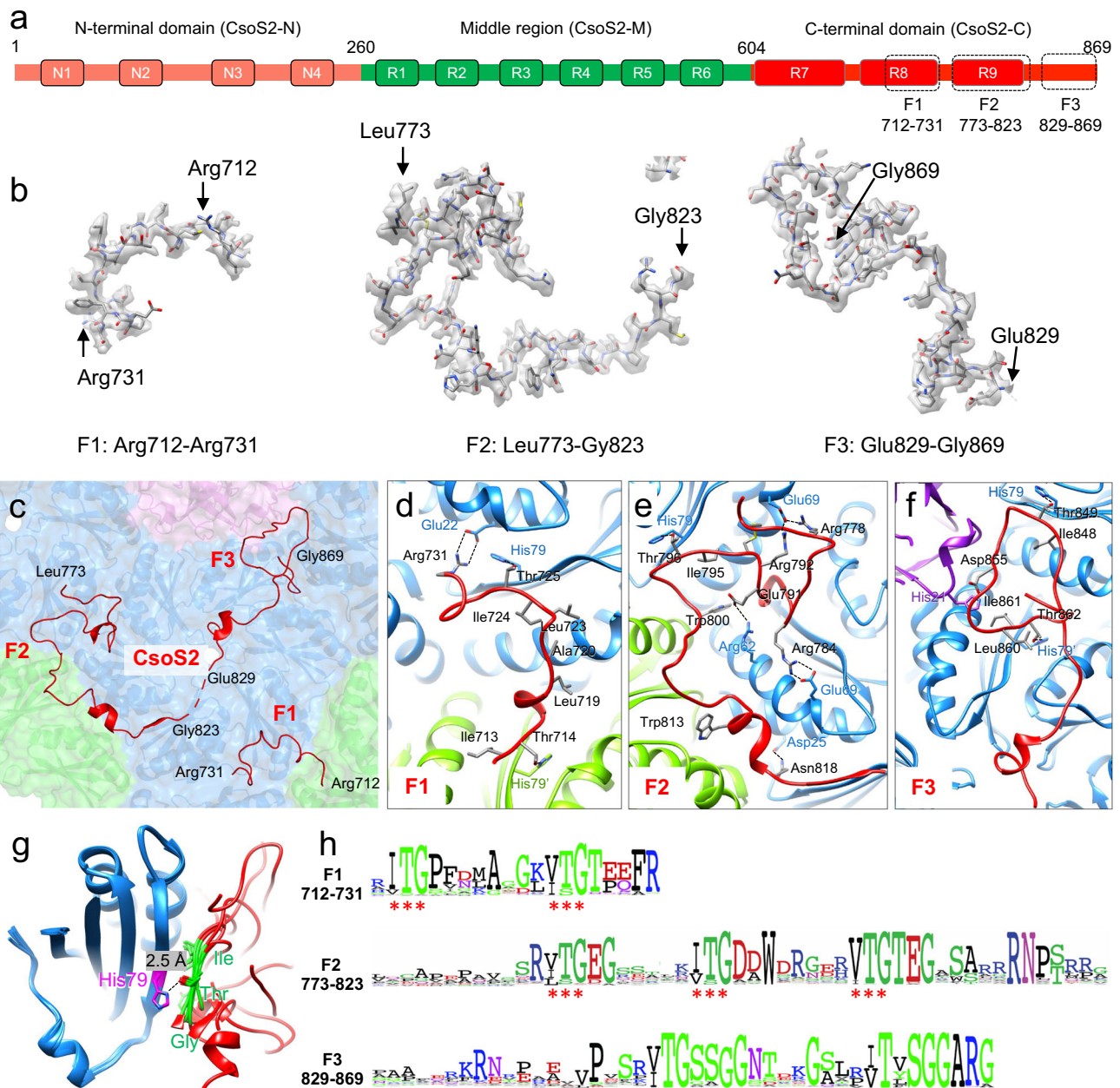

**Fig. 3 | CsoS2 binds to the shell through multivalent interactions with shell proteins and highly conserved interfaces via novel [IV]TG repeats. a** The domain arrangement of CsoS2. The N-terminal, Middle and C-terminal domains are colored in pink, green and red, respectively. Three dashed boxes indicate the structured fragments resolved in $T = 9$ shell. **b** CryoEM densities of F1-F3 with atomic models. **c** CsoS2 interactions with shell components, viewed from inside. Three structured fragments in the C-terminal domain, F1, F2 and F3, are identified and labelled. **d**–**f** Interaction interfaces between CsoS2 F1 (**d**), F2 (**e**) and F3 (**f**) fragments with shell components, CsoS1A (blue/green) and CsoS4A (purple). **g** Alignment of CsoS2-CsoS1A interacting motifs, showing the CsoS2 [IV]TG motif (green) in contact with CsoS1A His79. **h** Consensus sequences of CsoS2 C-terminal F1, F2 and F3 fragments from 100 CsoS2 sequences, plotted with Weblog. Asterisks indicate the conserved repeating [IV]TG motif present in each fragment.

incorporated into the isolated mini-shells, confirming the importance of the CsoS2 C-terminus in attaching to the shell.

The cryoEM structure of the $T = 9$ shell at 1.86 Å resolution enables the characterization of CsoS2 and its interactions with the shell at atomic details (Fig. 3b, Supplementary Fig. 3c). It is clearly well-resolved for the most of C-terminus in the cryoEM map (Fig. 3b), allowing unambiguous assignment of amino acid residues of three fragments of CsoS2-C: F1, Arg712−Arg731; F2, Leu773−Gly823; F3, Glu829−Gly869 (Fig. 3b). The rest of the CsoS2 regions were not resolved. The cryoEM structure revealed that the F1, F2, F3 fragments adopt completely different conformations from the AlphaFold2-

predicted structures (Supplementary Fig. 7d). They form extensive hydrogen bonds and salt bridges with both CsoS1A and CsoS4A at three distinct binding sites (Fig. 3d−f), with the surface contact area of 767.1, 3333.0, and 1851.7 Å$^2$, respectively. Such extensive, multivalent interactions strongly anchor CsoS2 to the shell inner surface (Fig. 3c).

Interestingly, all three CsoS2-C fragments, F1 to F3, bind specifically to the tri-capsomere interfaces (Fig. 3c). While CsoS2-F1 and F2 interact with three CsoS1A hexamers (Fig. 3d−f), CsoS2-F3 acts on the interface between one CsoS4A pentamer and two CsoS1A hexamers, with the extreme C-terminus buried inside the cavity formed by the interface (Fig. 3f). It is explicit that the CsoS2 C-terminus is completely

encapsulated inside the shell (Fig. 3c–f, Supplementary Fig. 7a), rather than being exposed to the exterior as previously proposed[28]. The three CsoS2-C F1-F3 fragments associate tightly with shell proteins simultaneously and essentially "stitches" the capsomeres together at the inner surface of the $T = 9$ shell. Therefore, CsoS2-C functions as a unique "molecular thread" reinforcing the connections between capsomeres thus potentially stabilizing the assembled α-carboxysome shell.

At the first glance, the three CsoS2 binding sites appear random (Fig. 3c–f). However, a close inspection of the interfaces uncovers a common interaction motif among these interactions: alignment of all the CsoS1A-CsoS2 interfaces identifies a repetitive Ile(Val)-Thr-Gly ([IV]TG) motif[28,29] that interacts with the β-strand of highly conserved CsoS1A through His79 and hydrogen bonds in the main chain (Fig. 3g, Supplementary Fig. 8). More intriguingly, the [IV]TG motif repeats itself about every 8 residues in all three CsoS2-C fragments (Supplementary Fig. 8b). Sequence alignment further reveals that the repeating [IV]TG motif is highly conserved in the CsoS2-family proteins (Fig. 3h, red asterisks), suggesting that the "molecular threading" of CsoS2-C is likely a widespread feature mediating α-carboxysome shell assembly. In addition to the repeating CsoS2 [IV]TG motif interacting with the shell proteins, there are many other specific interactions in all these interfaces such as the salt bridge between CsoS2 R731 and CsoS4A E22 (Fig. 3d), CsoS2 R784 and CsoS4A E69 (Fig. 3e) and hydrogen bond between CsoS2 R855 and CsoS4A H21 (Fig. 3f). In native α-carboxysomes, if CsoS2 must bind to the shell with F3 located to pentamer-hexamer interface, then the maximum CsoS2 occupancy would be 60 (12 pentamers). However, our recent study revealed that the native α-carboxysome from *Halothiobacillus* contains 863 hexamers and 192 CsoS2B as the full-length CsoS2[23], which indicates that CsoS2B does not have to restrict to all pentamer-hexamer interfaces. The local structures of CsoS2 on the authentic carboxysome shell could be more dynamic. Our structure therefore represents one of the local snapshots recapitulating the binding modes in native environments.

### CsoS2-C dictates α-carboxysome shell assembly and architecture

Since all the three CsoS2-C F1-F3 fragments form contacts with the shell proteins and contain multiple [IV]TG motifs (Fig. 3, Supplementary Fig. 8b–d), we further investigated the importance of individual CsoS2-C fragments and [IV]TG motifs to $T = 9$ shell formation. To this end, we designed four mini-shell constructs, mini-shell 4-6, with variations in the CsoS2-C (Supplementary Fig. 1a, b). Mini-shell 4-6 contain CsoS2-C fragment F1-F3 (S2-C1), F2-F3 (S2-C2) and F3 only (S2-C3), respectively (Fig. 4a, Supplementary Fig. 1a, b). In addition, a 7th mini-shell construct was generated where all CsoS2-C [IV]TG motifs were replaced by AAA (S2-Cm) in the background of mini-shell 4 (Fig. 4a). We measured ratio of assembled shell proteins to unassembled shell proteins by comparing the abundance of shell proteins in the soluble and the assembled forms. Interestingly, in the absence of CsoS2, mini-shell composed of CsoS4A and CsoS1A (mini-shell 1) assembles more efficiently than the mini-shell containing CsoS2-F3 fragment (mini-shell 6), but similar to the [IV]TG motif mutant (Cm). One plausible explanation is that the CsoS2-F3 may interfere shell assembly. It is also possible that CsoS4A-CsoS1A and S2-Cm assemble efficiently into different shell morphologies, such as $T = 4$ and $T = 3$ shells observed in mini-shell 1.

To systematically dissect the effects of CsoS2-C fragments on the morphology of shells, we further carried out cryoEM structural characterization of these four mini-shell variants containing S2-C1, C2, C3 and Cm. While CsoS2 (mini-shell 2) promotes assembly of large $T = 9$ shell (Fig. 1c), the mini-shell containing S2-C1 fragment, interestingly, leads to formation of shell with various sizes and symmetries: $T = 9$, $T = 7$, $T = 7\ Q = 6$, $T = 4$, $T = 4\ Q = 6$, and $T = 3$ (Fig. 4c, d, Supplementary Fig. 10, Supplementary Table 2). CsoS2-C fragments were

unambiguously identified in all the larger shells albeit with a relatively lower occupancy (Supplementary Fig. 11) compared with the full-length CsoS2 construct (Supplementary Fig. 10a–d), but not in the $T = 4$ and $T = 3$ shells. In contrast, the S2-C2 construct produces exclusively $T = 4$ shells (Fig. 4d), with little residual densities corresponding to the CsoS2 [IV]TG motifs (Supplementary Fig. 10e, f). The S2-C3 construct produced only $T = 4$ and $T = 3$ shells, both having a subpopulation missing pentamers (Fig. 4c, d). Notably, mutation of [IV]TG results in formation of only $T = 4$ and $T = 3$ shells (Fig. 4d), with the population ratio similar to the mini-shell 1 construct lacking CsoS2. These results illustrate the essential role of CsoS2-C and [IV]TG motif in controlling α-carboxysome shell assembly and overall architecture.

Carboxysomes are a paradigm of self-assembling proteinaceous organelles found in nature, offering compartmentalisation of enzymes and pathways to enhance carbon fixation. Given their significance in the global carbon cycle, carboxysomes are gaining increasing attention from fundamental studies and synthetic engineering, with the intent of generating metabolic factories for sustainably turbocharging carbon fixation and primary production. We devised a minimal system encompassing shell proteins and the linker CsoS2 to decipher the molecular principles of shell assembly and encapsulation. The distinctive multivalent interactions between CsoS2 C-terminus and shell proteins and between CsoS2 N-terminus and Rubisco are vital for governing the architectures of shell assemblies and encapsulation of Rubisco, respectively (Fig. 4e), while the actual role of CsoS2 middle region remains to be determined. Since the middle region also contains repetitive [IV]TG motif, it is likely that the middle region may also contribute to its interaction with shell components. CsoS2 may prefer to bind to these curved capsomer interfaces but would retain a certain affinity to flat interfaces, which is the case in the native carboxysomes facets. Advanced knowledge of carboxysome assembly will offer new strategies for design and engineering of carboxysome shell-based nanobioreactors and new cages in diverse biotechnological applications, such as enhancement of biocatalysis, food and energy production, molecule delivery, and therapeutics.

## Methods

### Generation of constructs
Primers (Supplementary Table 3) and the mutant *csoS2* sequence were ordered from Integrated DNA Technologies. The pHnCBS1D plasmid (Addgene, UK) was used as the template for amplification of native *csoS2, csoS4A*, and *csoS1A* genes from the *Halothiobacillus neapolitanus* genome and cloned into NcoI/XhoI and EcoRI linearized pBAD by Gibson assembly (New England Biolabs, UK)[51] to produce the synthetic mini-shell operons used in this study (Fig. 1b; Supplementary Fig. 1). Expression constructs were sequence verified by Eurofins Genomics and transformed into *E. coli* TOP10 for plasmid storage and expression.

### Isolation of α-carboxysome mini-shells
*E. coli* TOP10 cells containing the desired mini-shell construct were cultured in Lysogeny Broth (LB) supplemented with ampicillin (100 µgmL⁻¹) at 37 °C to an OD at 600 nm (OD$_{600}$) between 0.6 and 0.8. Mini-shell expression was then induced with 1 mM arabinose (Melford) at 22 °C for 20 h. Cells were harvested at $5000 \times g$ for 10 min and pellets resuspended in TEMB buffer (10 mM Tris-HCl pH = 8, 1 mM EDTA, 10 mM MgCl$_2$, 20 mM NaHCO$_3$) supplemented with 10% (v/v) CelLytic™ B cell Lysis Reagent (Sigma-Aldrich) and 0.1% Protease Inhibitor Cocktail (Sigma-Aldrich). Cell lysis was performed by sonication (MSE 8-75 MK2 sonicator, 6 cycles of 30 s ON/OFF) and cell debris removed by centrifugation at $27,000 \times g$, 30 min, 4 °C. The supernatant was subsequently loaded on top of 5 mL 30% (w/v) sucrose and mini-shells pelleted by ultracentrifugation at $250,000 \times g$, 16 h, 4 °C. By using a soft brush, the pelleted mini-shells were either resuspended in 1 mL TEMB for further isolation or in TEMB to the same volume as the supernatant to assess assembly ratio of the mini-shells.

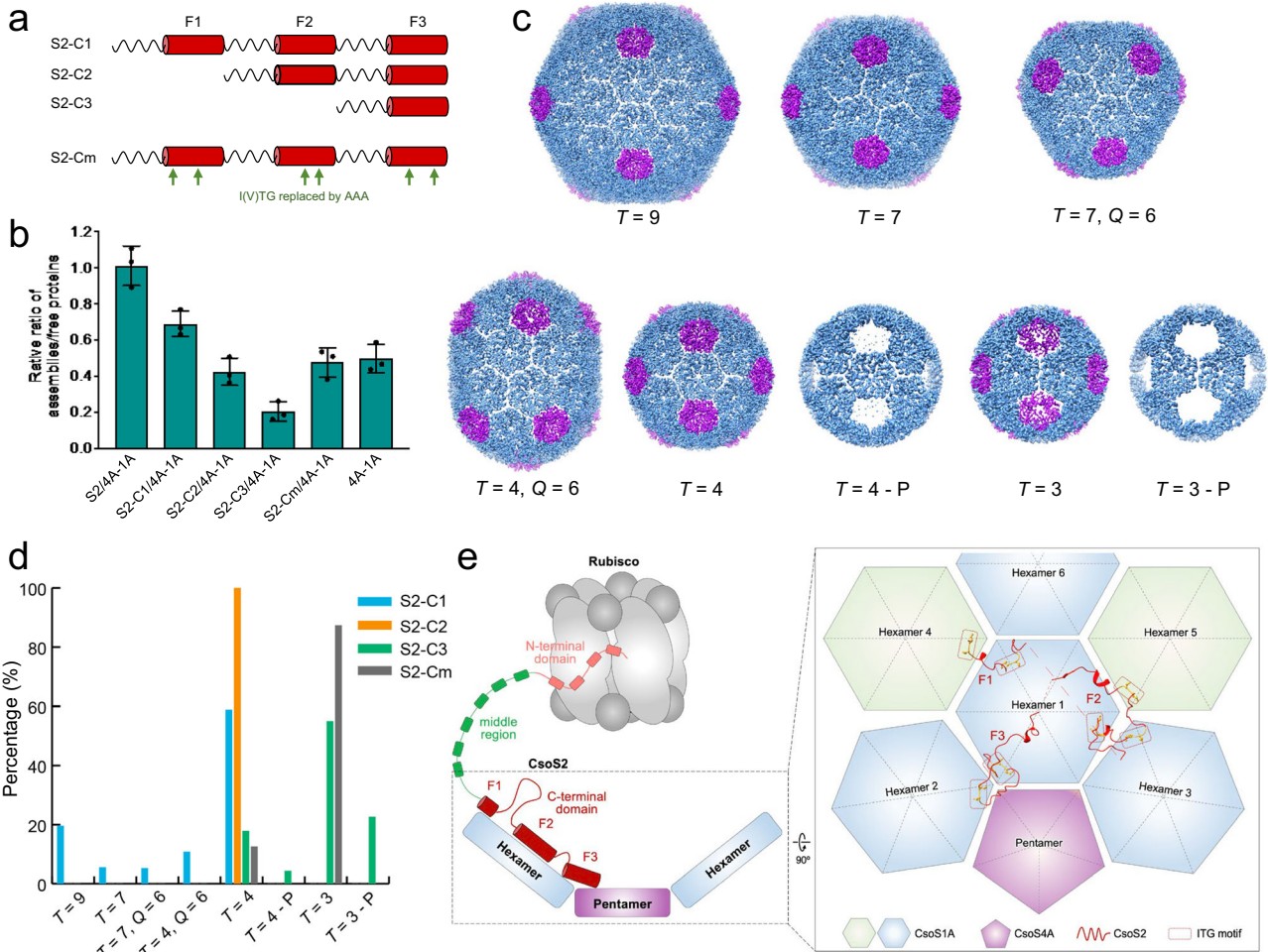

**Fig. 4 | CsoS2 C-terminal fragments interacts with shell proteins. a** Schematic of four CsoS2 constructs. **b** Quantification of shell assembly from different CsoS2 constructs. The ratios of assembled shell and free shell proteins are quantified from western blot experiments (*n* = 3, see also Fig. S9). The ratios are presented as mean values +/− SEM. 4A-1A denotes the CsoS4A-CsoS1A mini-shell. **c** Gallery of different shells formed in these constructs with hexamer in blue and pentamer in magenta. **d** Distribution of different shells in the four CsoS2 constructs as calculated from cryo-EM analysis; the number of particles for each shell is indicated in the Supplementary Tables 1 and 2. Source data are provided as a Source data file. **e** A schematic model of CsoS2 interacting with shell proteins and Rubisco using the C-terminal domain and N-terminal domain, respectively. Enlarged view shows the interactions of each fragment in CsoS2 C-terminal domain, F1, F2 and F3, with shell capsomers in the assembly.

For further purification of the mini-shells, resuspended pellets were centrifuged at 20,000 × *g*, 2 min, 4 °C. The supernatant was loaded onto a 10 mL step sucrose gradient consisting of 10%, 20%, 30%, 40% and 50% (w/v) sucrose and centrifuged at 70,000 × *g*, 16 h, 4 °C. Fractions were collected and analysed by SDS-PAGE. The fractions enriched with mini-shells were pooled and applied onto a HiTrap Q FF anion exchange chromatography column (Cytiva Life Sciences) equilibrated with buffer A (TEMB plus 50 mM NaCl). Mini-shells were eluted with a 0–40% linear gradient of buffer B (TEMB plus 1 M NaCl) and were found to be typically eluted at 30–35% buffer B. Purified samples were stored at 4 °C for further analysis.

### SDS-PAGE and immunoblot analysis
Protein samples mixed with SDS-PAGE loading buffer were heated at 99 °C for 10 min and electrophoresed on 15% SDS-PAGE gels. Gels were stained with Coomassie Brilliant Blue for SDS-PAGE analysis. For immunoblot analysis, gel electrophoresis was performed at 100 V 30 min, then 150 V 40 min, and blotted onto polyvinylidene fluoride membrane (Cytiva Life Sciences). Proteins were probed with the primary antibody anti-CsoS1A/B/C (Agrisera, Cat No. AS14 2760, dilution 1:5000), anti-CsoS2-N (1:10,000 dilution; synthesized by GenScript, NJ,

USA with the RGTRAVPPKPQSQG peptide), and anti-rabbit IgG secondary antibody (Agrisera, Cat No. AS09 602, dilution 1:10,000). Protein signals were analyzed using a Bio-Rad chemiluminescence kit (Bio-Rad, UK) and images were captured using ImageQuant LAS 4000 (GE Healthcare, USA).

### Dynamic light scattering (DLS) analysis
To measure the size distribution of isolated mini-shells and obtain an average size of their diameters, 1 mL mini-shell fractions (5–10 mg mL⁻¹) were analyzed by DLS using a ZetaSizer Nano ZS instrument (Malvern Panalytical Ltd, UK). All DLS measurements were performed in triplicate and Supplementary Fig. 2c was plotted with GraphPad Prism 9 (GraphPad Software, San Diego, California, USA).

### Negative-staining transmission electron microscopy
5 µL of isolated mini-shells were mounted on carbon grids (Carbon Films on 300 Mesh Grids Copper, Agar Scientific, UK) for 40 s, stained and washed with 60 µL of 2% uranyl acetate (Sigma-Aldrich), and excess stain wicked away with filter paper. Grids were left to air dry for at least 1 min. Images were recorded by the FEI Tecnai G2 Spirit Bio TWIN transmission electron microscope equipped with a Gatan Rio 16 camera. Images were visualized and analyzed by Fiji software[52]. Graphs

were created using GraphPad Prism 9 (GraphPad Software, San Diego, California USA, www.graphpad.com).

## CryoEM data collection

The cryoEM sample grids were prepared using Vitrobot. The grids (Quantifoil™ R 2/1 on 300 copper mesh) were glow-discharged using a plasma cleaner (Harrick Plasma) with the plasma level set to High position for 45 seconds using ambient air. 3 μL of mini-shell samples were applied to the grids and blotted with filter paper for 3 s before plunge freezing with liquid nitrogen-cooled ethane. The temperature was set to 20 °C and humidity at 100% during plunge freezing. The micrographs were taken using Thermo Scientific Titan Krios G3 microscope equipped with a Gatan K3 direct electron camera and BioQuantum energy filter (slit width 20 eV) or Falcon 4 with Selectris X energy filter (slit width 10 eV) using EPU (v2). Details of data collection parameters are listed in Supplementary Tables 1 and 2.

## CryoEM data processing

The data processing was performed using RELION (v3.1)[53] and cryoS-PARC (v3)[54]. The micrograph movies were gain normalized and motion corrected with MotionCor2 (v1.4.0)[55]. Contrast transfer function (CTF) was estimated using CtfFind4 (v4.1.14). In the first two dataset (Supplementary Table 1), three different sized shells ($T = 9$, $T = 4$ and $T = 3$) were observed on the raw micrographs and processed independently. The triangulation number (T) is the number of quasi-similar subunits per icosahedral asymmetric unit with possible value of T, and Q is an elongation number to describe the prolate/oblate icosahedra.

For the $T = 9$ particles, a subset of particles was picked manually in RELION to generate initial 2D class averages for auto-picking. Two rounds of 2D classification were performed, resulting a final dataset with 143,769 particles. 2D classification did not reveal ordered densities within the shell. An ab initio model was generated with I1 symmetry in RELION. 3D auto refine was carried out with the initial model reconstructed with I1 symmetry in RELION, which resulted in a density map with a mixed handedness. The resulted refined particles dataset was 3D classified into 10 classes, skipping alignment, which revealed two major classes, with opposite handedness. These two classes of particles were refined separately with per-particle CTF refinement and polishing. To combine the two classes, particles with opposite handedness were inverted by changing the refined Euler angle in RELION star file (phi and tilt). The two half maps were reconstructed using relion_reconstruct, with CTF and Eward sphere correction. The final combined density maps were masked, and B-factor sharpened (−47.66, automatically determined by RELION) with relion_postprocess, which resulted in a final map at 1.86 Å resolution (Supplementary Fig. 3 and Supplementary Table 1). Symmetry expansion and focused classification was carried out, which did not reveal alternative CsoS2 conformations binding to the shell proteins.

The small shells ($T = 4$ and $T = 3$) were processed in a similar way to $T = 9$ shell, except in particle picking step. A small number of small shell particles were manually picked in EMAN2.3 to train neural network, which was subsequently used to pick against the whole dataset. The coordinates of particles (EMAN2 box files) were imported into Relion for further processing in the same way as large shell. Similarly, a small portion of particles were found in opposite handedness after 3d refinement and classification, which was then corrected by updating the Euler angles in the Relion star file as above. The final map after per-particle CTF refinement and polishing is at 2.5 Å resolution.

The rest of datasets (S2-C1/4A-1A, S2-C2/4A-1A, S2-C3/4A-1A and S2-Cm/4A-1A) were processed in cryoSPARC (v3)[54]. Particles were picked by a combination of blob picking and template-based picking to ensure all the mini-shells on the micrographs are picked; the duplicated particles were removed after 2D classification. For S2-C1/4A −1A dataset, several new symmetries were identified, in addition to the $T = 9$, $T = 4$ and $T = 3$ shells. Initial 2D classification revealed particles

with diameter smaller than $T = 9$ shell but larger than $T = 4$ shell. These particles were divided into 4 different classes ($T = 7$, $T = 7$ $Q = 6$, $T = 4$ $Q = 6$, $T = 4$ $Q = 6$ class 2) after several rounds of 2D and 3D classification with C1 symmetry. Further 3D refinement with icosahedral symmetry ($T = 7$), D3 ($T = T$ $Q = 6$) and D5 ($T = 4$ $Q = 6$ and $T = 4$ $Q = 6$ class 2) symmetry were performed to obtain the final density map, respectively. The other three datasets were processed similarly with icosahedral symmetry applied during 3D classification and refinement. Data collection and classification results are summarized in Supplementary Table 2. The occupancy of CsoS2 in the $T = 9$ density maps is calculated using Occupy[56].

## Model building and refinement

Initial models were obtained from crystal structures of hexamer (PDB 2EWH) and pentamer (PDB 2RCF). For the $T = 9$ shell, the CsoS2 was traced manually into the density map in Coot (v0.8.9.2). At 1.86 Å resolution, the side chains of CsoS2 can be unambiguously placed (Fig. 3b). One asymmetric unit of the icosahedral shell with additional surrounding subunits were further refined in Phenix.refine (v1.19.2-4158)[57]. Water molecules were placed into density manually. The final icosahedral models were reconstructed in Chimera with symmetry command with I1 symmetry. Details of model geometry statistics are listed in Supplementary Table 1. Model alignment and comparison were performed in Chimera (v1.15)[58]. Figures were rendered in Chimera (v1.15)[58], ChimeraX (v1.5)[59] and Pymol (PyMOL Molecular Graphics System, Version 2.0 Schrödinger, LLC).

## Bioinformatics and structural prediction

Protein sequences assigned CsoS2, CsoS1A, and CsoS4A were blasted against the NCBI database of non-redundant protein sequences. A total of 395 sequences for CsoS2, 990 sequences for CsoS1A, and 970 sequences for CsoS4A were selected and aligned using Clustal Omega[60]. The resulting multiple sequence alignment files were used to determine conservation, visualised with WebLogo 3[61]. Phylogenetic tree was built using IQ-TREE web server[62] and visualized using iTOL 6.4.3[63]. The conservation score of CsoS2 fragments is calculated using ConSurf server (https://consurf.tau.ac.il/consurf_index.php). The structures of the CsoS2 N-terminus (1-260), middle region (261-604), and C-terminus (605-869) were predicted by AlphaFold2[64], accessed via ColabFold[65].

## Reporting summary

Further information on research design is available in the Nature Portfolio Reporting Summary linked to this article.

# Data availability

The data that support this study are available from the corresponding author upon request. The cryo-EM density maps and corresponding atomic models have been deposited in the EMDB and PDB, respectively. The accession codes are listed as follows: (1) CsoS4A and CsoS1A (mini-shell-1 construct): EMD-15798 and PDB 8B0Y [https://doi.org/10.2210/pdb8B0Y/pdb] for $T = 4$ shell, EMD-15792 for $T = 3$ shell; (2) Full-length CsoS2 with CsoS4A and CsoS1A (mini-shell-2 construct): EMD-15801 and PDB 8B12 [https://doi.org/10.2210/pdb8B12/pdb] for $T = 9$ shell, EMD-15799 and PDB 8B11 [https://doi.org/10.2210/pdb8B11/pdb] for $T = 4$ shell; The raw micrographs have been deposited to EMPIAR with accession code of EMPIAR-11559. (3) Truncated CsoS2 with F1-F3 fragments (mini-shell-4 construct): EMD-15722 ($T = 9$), EMD-15720 ($T = 7$), EMD-15595 ($T = 7$, $Q = 6$), EMD-15723 ($T = 4$, $Q = 6$ class 1), EMD-15724 ($T = 4$, $Q = 6$ class 2), EMD-15719 ($T = 4$); The raw micrographs have been deposited to EMPIAR with accession code of EMPIAR-11560. (4) Truncated CosS2 with F2-F3 fragments (mini-shell-5 construct): EMD-15611 for $T = 4$ shell; (5) Truncated CosS2 with F3 fragment (mini-shell-6 construct): EMD-15758 ($T = 4$ with pentamer), EMD-15759 ($T = 4$ without pentamer), EMD-15760 ($T = 4$ with pentamer)

and EMD-15761 ($T = 4$ without pentamer); (6) CsoS2-C with [IV]TG mutant: EMD-15834 ($T = 4$) and EMD-15762 ($T = 3$). Source data are provided with this paper.

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

## Acknowledgements

We acknowledge Diamond for access and support of the cryo-EM facilities at the U.K. national eBIC (proposals NT29812 and BI28713), funded by the Wellcome Trust, MRC, and BBSRC. We are grateful to Yuewen Sheng and Yun Song for their technical support in cryo-EM data collection at eBIC (Electron Bio-imaging Centre), Diamond Light Source, and Prof. Ian Prior and Mrs. Alison Beckett at the Liverpool Biomedical Electron Microscopy Unit for technical support of electron microscopy, and the Materials Innovation Factory (MIF) for provision of analytical equipment. This research was funded by the National Key R&D Program of China (2021YFA0909600, L-N.L.), the UK Wellcome Trust Investigator Award (206422/Z/17/Z, P.Z.), the National Natural Science Foundation of China (32070109, L-N.L.), the UK Biotechnology and Biological Sciences Research Council grants (BB/S003339/1, BB/V009729/1 and BB/R003890/1, L-N.L.), the ERC AdG grant (101021133, P.Z.), the Royal Society (URF\R\180030, RGF\EA\181061 and RGF\EA\180233, L-N.L.), and the Leverhulme Trust (RPG-2021-286, L-N.L.). The research was supported by the Wellcome Trust Core Award Grant Number 203141/Z/16/Z with additional support from the NIHR Oxford BRC. Q.J. was supported by a Liverpool-Chinese Scholarship Council PhD Studentship. P.C.N. was supported by a BBSRC DTP PhD Studentship.

## Author contributions

T.N., Q.J., L-N.L., and P.Z. conceived the project and designed the experiments; Q.J., P.C.N. G.F.D., and F.H. performed shell construction and characterization, as well as negative-staining EM; T.N. prepared cryoEM samples; T.N. J.R. and Y.Z. collected cryoEM data; T.N. processed cryoEM data and refined the structures with the assistance of H.D. and J.S.; T.N., L-N.L. and P.Z. analysed the structures. T.N., L-N.L., and P.Z. wrote the manuscript with support from all authors.

## Competing interests

The authors declare no competing interests.
