## [Peer Review File · Nature Communications]

Intrinsically disordered CsoS2 acts as a general molecular thread for α -carboxysome shell assemblyREVIEWER COMMENTS

Reviewer #1 (Remarks to the Author):

In this paper Ni et al describe the structural importance of CsoS2 in the formation of α -carboxysome. The authors show that CsoS2 is crucial to the formation of large carboxysome shells, probably by acting as a molecular thread. Furthermore, they uncover the binding mechanisms as well the relative importance of each CsoS2 motive to the formation of carboxysomes.

This work greatly adds to the current knowledge on α -carboxysome formation. While it was already known that CsoS2 is crucial to the making of shells, the exact reason and mechanism was unknown. Thereby the authors add a new glimpse of the importance of the disordered CsoS2 to carboxysome organisation and biogenesis. This may also, as the authors mention further guide ongoing efforts in engineering functionalized bacterial micro compartments.

While the experiments are thorough and the findings compelling, the article misses some discussion in the context of the existing literature in light of their findings. I have no objection to publication, but if they authors would clarify the few comments below this would add to the clarity of the article and define better their finding within the exiting literature.

Requests

- Lines 59-60 and 133-136: The authors mention that CsoS2 is vital to the making of carboxysome shells and cite literature that validates this statement. However in this work the authors obtain carboxysome shells in absence of CsoS2, while this could not be obtained in the native organisms or in heterologous systems (Cai et al, 2015; Long et al, 2018 and Li et al, 2020). Can the authors comment on this discrepancy in results? Is CsoS2 really necessary to make (functional) carboxysomes?

- Line 91 (and others): The parameters T and Q are not defined or explained in the article, this would be helpful to enable smooth reading of the article.

- Are shells of different T and Q found in nature? Relevant to know how important CsoS2 is to form functional carboxysomes

- Line 161 (paragraph and figure 3 and 4): How does permeability of the shell changes with incorporation of the linker? Could it be that CsoS2 partially impedes the permeability of the shell as it is partially occluding the central pore of CsoS1A?

Comments on the figures

Fig 1

- In panel b: please add the scheme of minishell 3 for completeness

Fig 4

- Panel b is badly formatted and it is unclear what "4A-1A" is

Methods

The methodology is sound and, though concise, sufficiently explained. I only have a few remarks on the SPA analysis:

- Line 337: Please add which CryoSparrc version was used
- Line 347: Could you clarify why a mixed handedness was obtained in the initial model, instead of just the right or wrong handedness
- Line 247: 3d -> 3D

References

1. Cai, F. et al. (2015). Advances in understanding carboxysome assembly in *Prochlorococcus* and *Synechococcus* implicate CsoS2 as a critical component. *Life* 5.
2. Long, B. M. et al. (2018) Carboxysome encapsulation of the CO₂-fixing enzyme Rubisco in tobacco chloroplasts. *Nat. Commun.* 9,.
3. Li, T. et al. (2020) Reprogramming bacterial protein organelles as a nanoreactor for hydrogen production. *Nat. Commun.* 11

Reviewer #2 (Remarks to the Author):

The authors present a highly interesting and well written manuscript investigating the role of CsoS2 in carboxysome assembly. The experimental results are of high quality and nicely presented and

conclusions are supported by the data. This manuscript is clearly impactful as the results elucidate the interaction sites of CsoS2 on the carboxysome shell and show specific molecular interactions mediated by specific motifs on the CsoS2 protein.

There are a number of specific comments to be addressed below, which I would expect the authors to attend to before the manuscript is accepted for publication.

Regarding the processing of the cryo-EM data – the CsoS2 binding sites seen in the T=7 shell structure appear to be well resolved – could the authors comment on how they relate to the asymmetric unit – it appears that they are only visible on one of the CsoS1A hexamers in the asymmetric unit – it is plausible that the CsoS2 protein would/could bind to other hexamers in the asymmetric unit, but this information would be lost in the icosahedral averaging in favour of the interaction seen if it dominates. Did the authors process the data with no symmetry imposed, and can they see any hints of alternative binding at other hexamers?

Data availability – in line with best practice in the cryo-EM community I would ask the authors to deposit their micrographs/particle sets in EMPIAR and link these to the EMDB entries.

Specific comments

Abstract

Line 24: 'Designed' – change word – in nature nothing is 'designed'

Line 33: CsoS2-goven- change to govern

Introduction

Line 40: 'past decade' – I would argue that our knowledge of bacterial organelles has a longer span than a decade. Admittedly, our molecular knowledge is recent, but knowledge of the carboxysome is 50+ years old.

Line 44: CO₂ fixation in all – change for reading ease: CO₂ fixation found in all...

Line 55: 'both containing multiple paralogous proteins' – clarify what you mean by both?

Line 65: 'in new contents': should this be 'in new contexts'

Results

Line 110: Clarify what the RMSDs are calculated against: I assume C α ?

Line 153: Clarify how the CsoS2 densities fit in the context of the icosahedral symmetry averaging used?

Line 174: ‘seemingly disordered interactions’: These interactions are clearly ‘ordered’ by virtue of their observability in the cryo-EM maps, but the protein does generally have limited secondary structural order. Please reword for clarity.

Line 175 and beyond: annotation of the I(V)TG motif – in keeping with community standards for motif annotation I think you should annotate this motif as [IV]TG i.e. either I or V and then T, G (see here for more details on sequence motif annotation: https://en.wikipedia.org/wiki/Sequence_motif)

Line 205: Define how the occupancy of the CsoS2 was calculated

Line 216: ‘significance in global carbon cycle’ – should be ‘significance in the global carbon cycle’

Methods

Line 289 – specify make/model of sonicator used.

Line 324 – ‘graphs were created’- specify which graphs/plots were produced in Prism

Line 332 – Please explicitly state which microscopes were used for data collection.

Line 345 – ‘ab initial’ should be ‘ab initio’

Line 347 – It isn’t clear here how Relion could produce a model in 3D refinement with mixed handedness – it presumably wouldn’t give a stable refinement and would not converge if this was the case. Could it be possible here that multiple datasets were merged and the models/maps were in opposite hands? It is also possible that T=7 capsids can have levo/dextro handedness – but these different hands cannot be superimposed as they occur at a level above individual protein handedness. Please clarify this and state the handedness of the T=7 capsids.

Figures/tables

Figure 3 legend – motif labelling as noted above.

Reviewer #3 (Remarks to the Author):

In the manuscript “Intrinsically disordered CsoS2 acts as a universal molecular thread for α -carboxysome assembly”, the authors report multiple variant cryo-EM structures of synthetic minimal alpha-carboxysomes (CsoS1A, CsoS4A, and CsoS2) with resolvable CsoS2 interactions with the interior of the minimal BMC shell.

Overall, this study is technically solid and interesting but overstates the conclusions that can be drawn from the reported experiments. The title of the paper implies that alpha-carboxysome assembly was studied, however, the catalytic components of alpha-carboxysomes are of course missing (Rubisco and carbonic anhydrase). What was studied was the assembly of a minimal shell in the presence of CsoS2. I

think the authors have to at least include Rubisco into their minimal system to make claims as broad as they are stated in the manuscript. The authors have to make it very clear that what they are working with is a non-native minimal shell system. The CsoS2 interactions in native alpha-carboxysomes might be substantially different. How difficult it is to draw general conclusions from a synthetic minimal system is highlighted by how easily shell size/symmetry can be modulated by CsoS2 mutants. The presence of Rubisco/CA might of course also heavily influence how CsoS2 interacts with the shell and what shell size/symmetry will be observed. I would recommend to substantially narrow the scope of the conclusions discussed in the paper before publication in Nature Communications.

Reviewer #4 (Remarks to the Author):

Key results

In their study titled “Intrinsically disordered CsoS2 acts as a universal molecular thread for α -carboxysome assembly”, Ni et al reconstituted minimal alpha-carboxysome shells containing the scaffolding protein CsoS2 and used cryo-EM to characterize their structure and architecture. To this end, they constructed a synthetic operon containing only the genes coding for the carboxysome shell proteins CsoS4A and CsoS1A from *Halothiobacillus neapolitanus*, and another operon containing these two genes plus the one coding for the CsoS2 protein. CsoS2 is a long (~869 residues), disordered protein that binds both shell and cargo proteins, and thus plays the role of molecular adaptor tethering cargo proteins to shell proteins during carboxysome assembly. CsoS2 is known to be essential, but the interaction between the shell proteins and CsoS2, as well as its mechanistic involvement in carboxysome assembly, remained poorly understood. In this study, Ni et al show that minimal carboxysome shells made up of only CsoS4A and CsoS1A form smaller shells and of different icosahedral symmetry compared to minimal shells made up of CsoS4A, CsoS1A and CsoS2. They discovered that the C-terminal region of CsoS2 very stably interacts with the shell proteins, “stitching” together hexameric and pentameric capsomeres. Given the conservation of CsoS2, they speculate that this interaction with shell proteins and mode of shell assembly may be widespread among bacterial species harboring alpha-carboxysomes.

Overall, the authors’ conclusions are supported by the data they present and the authors are careful to not overinterpret their results. Their approach is sound: determining 3D structures is one of the best ways to characterize and compare different icosahedral shell assemblies. The data are of sufficient quality to support the conclusions. To summarize, this study provides an important step towards a detailed biochemical understanding of carboxysome assembly and we enjoyed reading it! There are, however, a number of meaningful discussion points, some raw-data not submitted and some important control experiments missing.

Suggested improvements and comments to the authors

Major comments:

Results and Discussion

1. Assembly and structures of recombinant α -carboxysome shells

In this section, our main concern is the resolution claim of 1.86 Å for the mini-shell 2 T=9 structure. It may be correct, but is not consistent with the range of local resolutions of 2.1 to 2.4 Å reported in Figure S3C: the reported global resolution falls out of this range, while it should fall within the range (which is the case for the other structures presented in Figure S3). We are not able to determine which number is incorrect. It is possible that a mistake was introduced when preparing the figure. Please check this carefully.

In addition to FSC curves and maps colored by local resolution, it would be helpful to show a histogram of local resolution values (this is produced by Relion in the PDF file output by LocalRes jobs).

3. The model of CsoS2 C-term and its fit to the experimental map would be better shown in one of the main figures, as this is the main discovery being reported, and this visual is very compelling! We suggest moving the figure panel S7c (or a variant of it) to a main figure.

4. Lines 166-170: regarding the statement “The three CsoS2-C F1-F3 fragments associate tightly with shell proteins simultaneously and essentially “stitches” the capsomeres together at the inner surface of the T = 9 shell. Therefore, CsoS2-C functions as a unique “molecular thread” reinforcing the connections between capsomeres thus stabilizing the assembled α -carboxysome shell.”

It would be interesting to discuss and compare the artificial mini-shells analyzed in this study with native carboxysomes - which are substantially larger. In other words, in native carboxysomes, is every capsomere bound by a CsoS2 F1, F2 or F3 motif? For example, the in vivo stoichiometry of CsoS2B to shell capsomere is known (nicely determined by the same group). How many shell binding sites are there compared to S2 binding motifs? How far can one CsoS2 reach? Native carboxysomes have larger facets, do all CsoS2 interact with pentamers?

5. CsoS2-C dictates α -carboxysome shell assembly and architecture:

Regarding the shells missing pentameres: what could cause this exclusion of the pentamers from the assembly? This is not necessarily something to address experimentally, but it would be interesting to discuss.

6. Regarding the western blot of CsoS1A in the pellet and supernatant to assess the fraction of protein assembled in shells (Fig 4b and Fig. S9). What is the reason for substantially higher expression of CsoS1A in S2-C2/4A-1A (and S2-C2/4A-1A)? Judging from the western-blot all constructs form essentially the same amount of mini-shells, and the relative ratio of assembly seems to mostly correlate with CsoS1A expression. Does probing for the other components yield the same results, i.e. do expression levels of the S2-variants correlate with CsoS1A expression? or is S2 expressed in the same amount in all constructs and S2 concentration is what dictates the amount of shell in the pellet? In other words, to draw conclusions regarding assembly efficiency between the different constructs it is important to demonstrate that S2 and CsoS1A are expressed in the same ratio for all constructs. One would expect a similar result with all components.

7. Has His79 on S1B (and S4B) been mutated to see how this affects assembly? Except for the hydrogen bond between His79 and S2-Tyr, are there any other important types of molecular interactions that contribute to the binding?

Further, are there any theories of why F3 specifically interacts with the pentamer-hexamer interface while F1 and F2 do not?

8. CsoS2 exists as two isoforms transcribed from the same gene: CsoS2B - the full-length protein and CsoS2A - containing only the middle region and the N-terminal region and not the C-terminal shell binding motifs. It is unclear to us if CsoS2B/A or only CsoS2B has been used. CsoS2B would likely be the best for comparing mini-shell 2 with S2-C1, S2-C2 and S2-C2 constructs. If expressing both CsoS2B and 2A in mini-shell 2, what was the rationale for this? And how could this affect assembly? (By introducing a mutation removing the frameshift in the WT sequence one can produce the CsoS2B form. Chaijarasphong et al., 2016)

9. Related to point 10, a discussion about the role of the middle region in the assembly would be interesting. As is pointed out, this region contains the same motif as the C-terminal - can the middle region also interact with the shell? AlphaFold prediction for CsoS2 shows many secondary structures (albeit with various levels of confidence). Was there any indication that some regions of CsoS2 other than the C-ter could be ordered in these shells? Do the 2D class averages of the mini-shell 2 T=9 show any density (even fuzzy) inside the shell? Or is the rest of CsoS2 completely disordered? We understand that experiments on this are out of the scope of the paper, this is more of a curiosity and short discussion would be nice to add.

10. The author should carefully check their references to make sure there is a sound balance between self-citation and citing others' work. As it is now 12 out of 18 carboxysome references are self-citations. (counting refs. 1-30, as 31-44 are method refs.). Relevant papers from other groups are missing, and in some cases reviews are cited where an original reference would be more suitable.

Methods

11. CryoEM data collection

Please specify the glow discharge parameters: pressure, which gas was used (likely ambient air, but specify), current and duration.

Please fully specify the type of grids: were they gold or copper grids? What was the mesh size?

12. CryoEM data processing: Please specify the exact versions of all software programs used:

- RELION
- cryoSPARC
- MotionCor2
- CTFFIND4

13, Line 353: specify the value of the sharpening B-factor used for post-processing.

14. Model building and refinement: Please specify exact versions of all software programs used:

- Coot
- Phenix.refine (version of the phenix suite)
- Chimera
- ChimeraX

15. EMDB/PDB validation reports: Please consider depositing half maps and masks used for refinement and FSC calculation for all 15 structures deposited to the EMDB. They are required to calculate the masked FSC curve and thus to independently verify the resolution claims. Half maps and masks are also important for software and methods developers, so it benefits the whole cryo-EM community when they are deposited. Unfortunately, the EMDB does not make this a strict requirement.

Of the 15 EMDB validation reports provided, only two are marked as “for manuscript review” (EMD-15595 and EMD-15611), while the others are draft validation reports with many fields left empty. While

this does not raise much doubt about the validity of the data, it also prevents a complete examination. Please submit the final EMDB validation reports.

Finally, we encourage the authors to deposit their raw movies into the EMPIAR database, at least for the datasets that yielded the 3 structures with an atomic model deposited in the PDB. We do not consider this a strict requirement for publication, but wish to promote this practice because it is highly beneficial for the cryo-EM community at large.

Minor comments

Line 27: "Its mechanism of action, however, is not known". This statement is not entirely true. Modify so it is clear that the mechanism is partly known e.g. C-term interacts with shell, N-term with Rubisco and details of N-term-Rubisco interactions are known.

Line 30: "the intrinsically disordered CsoS2 C-terminus is well-structured" sounds like a contradictory statement. Maybe try to rephrase.

Line 40: we suggest adding biochemistry to this list of methods.

Line 41: add that BMCs are composed entirely of proteins.

Line 59: "shell first" could be misinterpreted to believe that cargo enters the BMC after the shell is formed. Consider rephrasing this.

Line 83: specify which form of CsoS2 was used, see comment 10.

Lines 92-93: "A medium sized shell (~23 nm, T = 4) are present" should be edited to either "Medium sized shells [...] are present" or "A medium sized shell [...] is present".

Line 101: "extra densities not accounted for CsoS1A and CsoS4A" should be edited to "extra densities not accounted for by CsoS1A and CsoS4A".

Line 136: "CosS2" is a typo, should be "CsoS2".

Line 139: change "arrangement" to "sequence" or "short linear motifs (SLiMs)"

Line 169: do we know that the assembled shells are stabilized by S2 binding? The manuscript shows S2 involvement in the assembly process but does not provide evidence of higher stability of assembled shells. Consider rephrasing this. Same comment applies to Fig 3 legend.

Lines 179-180: the statement that "the molecular threading of CsoS2-C is likely a universal feature" is a bold claim. Unless the authors can present data to support the "universal" character of this claim, we suggest rephrasing as follows (or similar, at the discretion of the authors): "the molecular threading of CsoS2-C is likely a widespread feature".

Line 283: were TOP10 cells used for expression or is this a typo?

Line 291: "cell pellets" should be "pelleted mini-shells" (or similar)

Lines 300-301: it would be helpful for readers to specify at roughly which salt concentration the mini-shells typically elute.

Line 319: Change " Five" to 5

Line 328: Change "Three" to 3

Line 332: "a Gatan K3 director electron camera". There is a typo.

Line 345: "An ab initial model". Typo, this should read "an ab initio model".

Line 379: "Initial models from crystal structures of hexamer (PDB 2EWH) and pentamer (PDB 2RCF)." This sentence needs a verb.

Line 380: Define which are the large shells.

Figure 1a: split CsoS2 at $\frac{2}{3}$ (instead of $\frac{1}{2}$) as this better represent the size of S2B vs S2A

Figure 1b: specify if S2B or S2B/A was used for mini-shells

Figure S2a: due to other impurities on the gel can the band for CsoS2A be accurately assigned? If CsoS2A gets inside the carboxysome in mini-shell 2 this implies that other parts than the C-term of CsoS2 interacts with the shell or that CsoS2 self-interacts. If this is the case this should be commented on. See also comment 10.

Figure 2b, c: maybe use a different color scheme. Blue and orange are easy to distinguish, but gray is difficult to see.

Figure 3c-e: add hydrogen bond between His79 and S2-Tyr. For easier visualization we suggest adding which fragment is shown in each panel instead of only in the figure legend.

Figure 3f: add distance between His79 and S2-Tyr

Figure 4d: y-label is missing. what were the total numbers of shells that were counted to determine these proportions?

Figure 4e: a small comment: it would be nice if this cartoon figure was somewhat in scale. Now S2 middle region and N-term are short in comparison to C-term and Rubisco is very big.

Figure S3a-b: Have these micrographs accidentally been switched?

Figure S6: Which species are included in the clade named gammaproteobacteria? This is a class of bacteria while the other clades which are marked are grouped according to genus. Can this be a typo? Note that Halothiobacillus is a gammaproteobacteria.

Figure S7d: check that the RMSD values are correct. It looks as if 0.6 and 1.3 are swapped.

Figure S8c: how was the conservation score calculated? Using the ConSurf server?

Figure S10: maybe measure occupancy of CsoS2 in these maps with Occupy? See <https://doi.org/10.1101/2023.01.18.524529> and <https://occupy.readthedocs.io/en/latest>

This would enable a quantitative assessment of whether CsoS2 is present or not, in addition to the visual assessment provided by the figure.

Figure S8a: include distances of the hydrogen bonds. Which S2 fragment is shown here?

Bioinformatics: it is good practice to include a SI table with all sequences used for building MSAs and phylogenetic trees.

Point-to-point responses to reviewers' comments

Reviewer #1 (Remarks to the Author):

In this paper Ni et al describe the structural importance of CsoS2 in the formation of α -carboxysome. The authors show that CsoS2 is crucial to the formation of large carboxysome shells, probably by acting as a molecular thread. Furthermore, they uncover the binding mechanisms as well the relative importance of each CsoS2 motive to the formation of carboxysomes.

This work greatly adds to the current knowledge on α -carboxysome formation. While it was already known that CsoS2 is crucial to the making of shells, the exact reason and mechanism was unknown. Thereby the authors add a new glimpse of the importance of the disordered CsoS2 to carboxysome organisation and biogenesis. This may also, as the authors mention further guide ongoing efforts in engineering functionalized bacterial micro compartments. While the experiments are thorough and the findings compelling, the article misses some discussion in the context of the existing literature in light of their findings. I have no objection to publication, but if they authors would clarify the few comments below this would add to the clarity of the article and define better their finding within the exiting literature.

Requests

- Lines 59-60 and 133-136: The authors mention that CsoS2 is vital to the making of carboxysome shells and cite literature that validates this statement. However in this work the authors obtain carboxysome shells in absence of CsoS2, while this could not be obtained in the native organisms or in heterologous systems (Cai et al, 2015; Long et al, 2018 and Li et al, 2020). Can the authors comment on this discrepancy in results? Is CsoS2 really necessary to make (functional) carboxysomes?

CsoS2 is vital to make functional carboxysomes, serving at least as a linker protein connecting carboxysomes shell and Rubisco. In this work, small synthetic shell (T=4 and T=3) can be formed with the basic shell components CsoS1A and CsoS4A without CsoS2. However, CsoS2 is crucial for the formation of the larger shells (T=9), which likely more functional relevant despite that it is still smaller than the native carboxysomes (22-37 nm vs ~100nm), and the intact empty shells (Li et al., Nat Commun 2020). Overall, our results indicate the significance of CsoS2 in formation of large, complex α -carboxysome shells. We have made it clear in the revised manuscript.

- Line 91 (and others): The parameters T and Q are not defined or explained in the article, this would be helpful to enable smooth reading of the article.

We thank the reviewer's suggestion and add the explanation of T and Q in the Methods section: "The triangulation number (T) is the number of quasi-similar subunits per icosahedral asymmetric unit with possible value of T, and Q is an elongation number to describe the prolate/oblate icosahedra".

- Are shells of different T and Q found in nature? Relevant to know how important CsoS2 is to form functional carboxysomes

Natural carboxysomes are generally polyhedron with 20 faces and 12 vertices, but highly polymorphic, deviating from a canonical icosahedron (Ni et al., Nature Comm 2022). Each native carboxysome varies in shape and size, most likely associated with different T and Q numbers, thus the structure of nature carboxysomes have not been determined. The mini-shells that we engineered in this study based on a reductionist approach to understanding the fundamental phenomenon allowed us to derive high-resolution structures and decipher the molecular principles underlying the formation of the α -carboxysome shell.

- Line 161 (paragraph and figure 3 and 4): How does permeability of the shell changes with incorporation of the linker? Could it be that CsoS2 partially impedes the permeability of the shell as it is partially occluding the central pore of CsoS1A?

We thank the reviewer to raise this interesting topic. It is plausible that part of CsoS2 partially block the central pore of CsoS1A, which may impede the passage of small ions or substrate molecules across the shell. In addition, only one-third of hexamers in the T=9 mini-shell have clearly resolved CsoS2 occupying the center of CsoS1A hexamer. In native carboxysomes which are much larger and contain multiple shell paralogs, the ratio of CsoS1A/C hexamer to CsoS2 is approximately 4:1 (863:192, Sun et al, mBio 2022), suggesting that CsoS2 may bind and affect hexamers other than the one connected to the pentamer. Furthermore, previous studies have revealed the importance of CsoS4 pentamers and CsoS1D (with a larger pore size) in shell permeability (Cai et al., PloS One 2009; Klein et al., J Mol Biol 2009). Therefore. the shell permeability is a complex matter.

Comments on the figures

Fig 1

- In panel b: please add the scheme of minishell 3 for completeness

The schematics for all the mini-shell constructs used in this study have been included in the Figure S1.

Fig 4

- Panel b is badly formatted and it is unclear what "4A-1A" is

We added description of 4A-1A in the figure legend: 4A-1A denotes CsoS4A-CsoS1A.

Methods

The methodology is sound and, though concise, sufficiently explained. I only have a few remarks on the SPA analysis:

- Line 337: Please add which CryoSPARC version was used

CryoSPARC (v3.3) was used.

- Line 347: Could you clarify why a mixed handedness was obtained in the initial model, instead of just the right or wrong handedness

We exported the same set of particles and re-processed the data starting from ab initio model generation to 3D refinement in cryoSPARC, which resulted a single handedness map, instead of a map with mixed handedness.

We have further discussed this issue in RELION with Sjors Scheres. He confers that both hands are equally valid from an image processing point of view, and suggests an initial model with a stronger hand (or higher resolution) to avoid convergence onto a mixture.

- Line 247: 3d -> 3D

Changed

References

1. Cai, F. et al. (2015). Advances in understanding carboxysome assembly in *Prochlorococcus* and *Synechococcus* implicate CsoS2 as a critical component. *Life* 5.
2. Long, B. M. et al. (2018) Carboxysome encapsulation of the CO₂-fixing enzyme Rubisco in tobacco chloroplasts. *Nat. Commun.* 9,.
3. Li, T. et al. (2020) Reprogramming bacterial protein organelles as a nanoreactor for hydrogen production. *Nat. Commun.* 11

References are fixed.

Reviewer #2 (Remarks to the Author):

The authors present a highly interesting and well written manuscript investigating the role of CsoS2 in carboxysome assembly. The experimental results are of high quality and nicely presented and conclusions are supported by the data. This manuscript is clearly impactful as the results elucidate the interaction sites of CsoS2 on the carboxysome shell and show specific molecular interactions mediated by specific motifs on the CsoS2 protein.

We thank the reviewer's positive comments.

There are a number of specific comments to be addressed below, which I would expect the authors to attend to before the manuscript is accepted for publication.

Regarding the processing of the cryo-EM data – the CsoS2 binding sites seen in the T=7 shell structure appear to be well resolved – could the authors comment on how they relate to the asymmetric unit – it appears that they are only visible on one of the CsoS1A hexamers in the asymmetric unit – it is plausible that the CsoS2 protein would/could bind to other hexamers in the asymmetric unit, but this information would be lost in the icosahedral averaging in favour of the interaction seen if it dominates. Did the authors process the data with no symmetry imposed, and can they see any hints of alternative binding at other hexamers?

This is an interesting point. The CsoS2 binding sites in the T = 9 shell are well resolved, whereas in other shells (such as $T = 7$, $T = 4$ $Q = 6$ and $T = 7$ $Q = 6$) these sites are partially resolved. We performed 3D refinement without icosahedral symmetry and could only obtain a very anisotropic density map, which is difficult for further interpretations (Figure R1). To further investigate this, we performed symmetry-expansion and focused classification of asymmetric unit from the icosahedral symmetry refinement, which revealed no alternative conformation of CsoS2.

Figure R1 | T=9 shell data analysis. (a) 2D class averages of T=9 shells. (b) 3D refinement of T=9 shell with and without icosahedral symmetry. (Top): a central area is zoomed in from both maps; (Bottom): A central slice of the density map showing no discernable feature inside the shell. The particle dataset was 2x binned to speed up the refinement and classification. Without icosahedral symmetry, the density map is very anisotropic. Scale bar = 20 nm.

The CsoS2 C-terminal fragments are only visible on one of three hexamers in the asymmetric unit. A possible explanation for such specificity is: in the T=9 shell, the F3 segment of CsoS2 is uniquely positioned at the pentamer-hexamer interface, which is different to the hexamer-hexamer interfaces that F1 and F2 bind to. The short amino acid linkage between F2 and F3 is likely to restrain the possible location of F2 with respect to F3 on the shell, thus form such particular arrangement. Furthermore, the measured hexamers/pentamers to CsoS2 ratio is about 4:1, supporting such configuration.

Data availability – in line with best practice in the cryo-EM community I would ask the authors to deposit their micrographs/particle sets in EMPIAR and link these to the EMDB entries.

We have begun uploading the raw data to EMPIAR. This was our plan nonetheless.

Specific comments

Abstract

Line 24: ‘Designed’ – change word – in nature nothing is ‘designed’

Changed to “found”.

Line 33: CsoS2-goven- change to govern

Changed.

Introduction

Line 40: 'past decade' – I would argue that our knowledge of bacterial organelles has a longer span than a decade. Admittedly, our molecular knowledge is recent, but knowledge of the carboxysome is 50+ years old.

Changed to “decades”.

Line 44: CO₂ fixation in all – change for reading ease: CO₂ fixation found in all...

Changed.

Line 55: 'both containing multiple paralogous proteins' – clarify what you mean by both?

Changed to “... both hexamers and pentamers containing multiple paralogous proteins...”.

Line 65: 'in new contents': should this be 'in new contexts'

Changed to: “...generating new caging nanomaterials in new contexts ...”

Results

Line 110: Clarify what the RMSDs are calculated against: I assume C α ?

Yes, the RMSDs are calculated based on C- α . A description of the comparison is inserted “...three icosahedral symmetries range 0.180-0.231 Å and 0.240-0.251 Å (calculated from the pairwise comparison of C α)...”.

Line 153: Clarify how the CsoS2 densities fit in the context of the icosahedral symmetry averaging used?

The density map is with the icosahedral symmetry averaging, with each asymmetric unit containing 1 CsoS2 molecule. Therefore, no further symmetry was applied to CsoS2. The three fragments were resolved to high resolution, suggesting that these fragments of CsoS2 follows icosahedral symmetry in the shell, while other ScoS2 regions may not.

From the 1.86Å resolution density map, we can unambiguously fit the three fragments of the CsoS2. Of note, there are some weak densities connecting F2 and F3 fragment, correlating with the linker length between F2 and F3 fragment. Therefore, the F2 and F3 in the asymmetric unit are confirmed to belong to the same polypeptide. However, there are 42 amino acid residues between F1 and F2. It is not possible to ascertain that the assigned F1 belong to the same polypeptide as F2 and F3. We provide an overview of CsoS2 polypeptides in the context of icosahedral symmetry (Figure R2).

Figure R2 | Structure of CsoS2 in the icosahedral symmetrized shell.

(a) Structure of T=9 shell visualized from internal surface. CsoS2 is colored in red. (b) CsoS2 fragments in the context of icosahedral symmetry. Three fragments are colored differently, F1 in green, F2 in orange and F3 in red.

Line 174: ‘seemingly disordered interactions’: These interactions are clearly ‘ordered’ by virtue of their observability in the cryo-EM maps, but the protein does generally have limited secondary structural order. Please reword for clarity.

The disordered or very flexible region become ordered once it binds to the partners, while other parts of the protein remain flexible and not observable by cryoEM averaging. We have reworded for clarity.

Line 175 and beyond: annotation of the I(V)TG motif – in keeping with community standards for motif annotation I think you should annotate this motif as [IV]TG i.e. either I or V and then T, G (see here for more details on sequence motif annotation:

https://en.wikipedia.org/wiki/Sequence_motif)

Thank you and we have annotated this motif as [IV]TG motif throughout the text.

Line 205: Define how the occupancy of the CsoS2 was calculated

We initially estimated the occupancy based on the density threshold and now included a quantitative measurement using Occupy (Figure R3, and new Figure S11).

Figure R3 | Quantification of CsoS2 occupancy by Occupy. (a) Occupancy map of T=9 shell from the full-length CsoS2 construct, with a F3 fragment zoomed in (right). (b) Occupancy map of T=9 shell from the truncated S2-C1 construct, with a F3 fragment zoomed in (right). Density map of a hexamer from the T=9 shell colored by occupancy from 0.4 to 1 (red to green). F3 fragment in T=9 S2-C1 construct has lower occupancy.

Line 216: ‘significance in global carbon cycle’ – should be ‘significance in the global carbon cycle’

Changed to ‘significance in the global carbon cycle’.

Methods

Line 289 – specify make/model of sonicator used.

The model of the sonicator that we used is MSE 8-75 MK2 sonicator.

Line 324 – ‘graphs were created’- specify which graphs/plots were produced in Prism

We have moved this sentence to ‘Dynamic Light Scattering analysis’ in Methods and reworded it as “All DLS measurements were performed in triplicate and Fig. S2B was plotted with GraphPad Prism 9 (GraphPad Software, San Diego, California, USA)”.

Line 332 – Please explicitly state which microscopes were used for data collection.

Data were collected using Titan Krios G3 microscopes: "... The micrographs were taken using Thermo Scientific Titan Krios G3 microscope."

Line 345 – ‘ab initial’ should be ‘ab initio’

Changed to “*ab initio*”.

Line 347 – It isn’t clear here how Relion could produce a model in 3D refinement with mixed handedness – it presumably wouldn’t give a stable refinement and would not converge if this was the case. Could it be possible here that multiple datasets were merged and the models/maps were in opposite hands? It is also possible that T=7 capsids can have levo/dextro handedness – but these different hands cannot be superimposed as they occur at a level above individual protein handedness. Please clarify this and state the handedness of the T=7 capsids.

We exported the same set of particles and re-processed the data starting from *ab initio* model generation to 3D refinement in cryoSPARC, which resulted a single handedness map, instead of a map with mixed handedness.

We have further discussed this issue in RELION with Sjors Scheres. He confers that both hands are equally valid from an image processing point of view and suggests an initial model with a stronger hand (or higher resolution) to avoid divergence onto a mixture.

Figures/tables

Figure 3 legend – motif labelling as noted above.

The motif labelling has been changed to [IV]TG throughout the text and figure legends.

Reviewer #3 (Remarks to the Author):

In the manuscript “Intrinsically disordered CsoS2 acts as a universal molecular thread for α -carboxysome assembly”, the authors report multiple variant cryo-EM structures of synthetic minimal alpha-carboxysomes (CsoS1A, CsoS4A, and CsoS2) with resolvable CsoS2 interactions with the interior of the minimal BMC shell.

Overall, this study is technically solid and interesting but overstates the conclusions that can be drawn from the reported experiments. The title of the paper implies that alpha-carboxysome assembly was studied, however, the catalytic components of alpha-carboxysomes are of course missing (Rubisco and carbonic anhydrase). What was studied was the assembly of a minimal shell in the presence of CsoS2. I think the authors have to at least include Rubisco into their minimal system to make claims as broad as they are stated in the manuscript. The authors have to make it very clear that what they are working with is a non-native minimal shell system. The CsoS2 interactions in native alpha-carboxysomes might be substantially different. How difficult it is to draw general conclusions from a synthetic minimal system is highlighted by how easily shell size/symmetry can be modulated by CsoS2 mutants. The presence of Rubisco/CA might of course also heavily influence how CsoS2 interacts with the shell and what shell size/symmetry will be observed. I would recommend to substantially narrow the scope of the conclusions discussed in the paper before publication in Nature Communications.

We appreciate the reviewer's comments. We modified the title to "Intrinsically disordered CsoS2 acts as a general molecular thread for α -carboxysome shell assembly"

Native carboxysomes are highly variable and polymorphic, not amenable for high resolution structural analysis. We took a reductionist approach to understanding the fundamental phenomenon, and engineered the mini shells using "bottom-up" approaches which allowed us to derive high-resolution structures and understand the principle of shell formation. We would argue that such an approach is typical in the field of structural biology and synthetic biology and has been vastly successful. To determine high-resolution structures of protein complexes and assemblies we generally use *in vitro* systems such as bacterial expression for eukaryotic proteins, highly purified and isolated from its native environment and other interacting partners, sometimes detergent solubilized for membrane proteins, crystallized under artificial buffer conditions and at non-physiological concentrations. Of course, we need to bear in mind that the native system is likely behave differently due to its complexity. We recognize the importance of studying the biological systems in their native context. In fact, we are one of the leading groups in the emerging field of *in situ* structural biology.

We have made it clear that this is an engineered mini-shell system in the revised manuscript.

Reviewer #4 (Remarks to the Author):

Key results

In their study titled "Intrinsically disordered CsoS2 acts as a universal molecular thread for α -carboxysome assembly", Ni et al reconstituted minimal alpha-carboxysome shells containing the scaffolding protein CsoS2 and used cryo-EM to characterize their structure and architecture. To this end, they constructed a synthetic operon containing only the genes coding for the carboxysome shell proteins CsoS4A and CsoS1A from *Halothiobacillus neapolitanus*, and another operon containing these two genes plus the one coding for the CsoS2 protein. CsoS2 is a long (~869 residues), disordered protein that binds both shell and cargo proteins, and thus plays the role of molecular adaptor tethering cargo proteins to shell proteins during carboxysome assembly. CsoS2 is known to be essential, but the interaction between the shell proteins and CsoS2, as well as its mechanistic involvement in carboxysome assembly, remained poorly understood. In this study, Ni et al show that minimal carboxysome shells made up of only CsoS4A and CsoS1A form smaller shells and of different icosahedral symmetry compared to minimal shells made up of CsoS4A, CsoS1A and CsoS2. They discovered that the C-terminal region of CsoS2 very stably interacts with the shell proteins, "stitching" together hexameric and pentameric capsomeres. Given the conservation of CsoS2, they speculate that this interaction with shell proteins and mode of shell assembly may be widespread among bacterial species harboring alpha-carboxysomes.

Overall, the authors' conclusions are supported by the data they present and the authors are careful to not overinterpret their results. Their approach is sound: determining 3D structures is one of the best ways to characterize and compare different icosahedral shell assemblies. The data are of sufficient quality to support the conclusions. To summarize, this study provides an important step towards a detailed biochemical understanding of carboxysome assembly and we enjoyed reading it! There are, however, a number of meaningful discussion points, some raw-data not submitted and some important control experiments missing.

Suggested improvements and comments to the authors

Major comments:

Results and Discussion

1. Assembly and structures of recombinant α -carboxysome shells

In this section, our main concern is the resolution claim of 1.86 Å for the mini-shell 2 T=9 structure. It may be correct, but is not consistent with the range of local resolutions of 2.1 to 2.4 Å reported in Figure S3C: the reported global resolution falls out of this range, while it should fall within the range (which is the case for the other structures presented in Figure S3). We are not able to determine which number is incorrect. It is possible that a mistake was introduced when preparing the figure. Please check this carefully.

We apologies for the mistake in the range of local resolution which was due to taking the wrong half maps for local resolution estimation. The overall resolution was calculated correctly, 1.86 Å. We have corrected the Figure S3c.

2. In addition to FSC curves and maps colored by local resolution, it would be helpful to show a histogram of local resolution values (this is produced by Relion in the PDF file output by LocalRes jobs).

We appreciate the reviewer's suggestion. We have also included the histogram of local resolution values for the three maps (Figure R4).

Figure R4 | Histograms of local resolution distribution for three shells, calculated from RELION.

3. The model of CsoS2 C-term and its fit to the experimental map would be better shown in one of the main figures, as this is the main discovery being reported, and this visual is very compelling! We suggest moving the figure panel S7c (or a variant of it) to a main figure.

We agree with the reviewer, this visual is indeed compelling. We have moved the figure panel S7c to the main Figure 3b.

4. Lines 166-170: regarding the statement “The three CsoS2-C F1-F3 fragments associate tightly with shell proteins simultaneously and essentially “stitches” the capsomeres together at the inner surface of the T = 9 shell. Therefore, CsoS2-C functions as a unique “molecular thread” reinforcing the connections between capsomeres thus stabilizing the assembled α -carboxysome shell.”

It would be interesting to discuss and compare the artificial mini-shells analyzed in this study with native carboxysomes - which are substantially larger. In other words, in native carboxysomes, is every capsomere bound by a CsoS2 F1, F2 or F3 motif? For example, the in vivo stoichiometry of CsoS2B to shell capsomere is known (nicely determined by the same group). How many shell binding sites are there compared to S2 binding motifs? How far can one CsoS2 reach? Native carboxysomes have larger facets, do all CsoS2 interact with pentamers?

In the native carboxysome, if CsoS2 must bind to the shell with F3 located to pentamer-hexamer interface, then the maximum CsoS2 occupancy is 60 (12 pentamers). Our recent study revealed that the native α -carboxysome from *Halothiobacillus* contains 863 hexamers and 192 CsoS2B as the full-length CsoS2 (Sun et al, mBio 2022), which indicates that CsoS2B does not have to restrict to all pentamer-hexamer interfaces. The local structures of CsoS2 on the authentic carboxysome shell could be more dynamic. Our structure likely represents stable binding modes in native environments.

Interestingly, in native carboxysomes as revealed by cryo-electron tomography study, the occupancy of CsoS2 in different locations can vary. Given that CsoS2 has its C-terminal region anchored to the carboxysome shell, the N-terminal region would reach the Rubiscos for packaging. Assuming that the middle region of CsoS2 is totally disordered, we estimate the distance between the N-terminal region and C-terminal region to be 70 nm (assuming 200 amino acid residues in CsoS2 are totally disordered: $200 \times 0.35\text{nm} = 70\text{ nm}$), which is sufficient for CsoS2 to bind to Rubiscos buried in the centre of carboxysomes. However, our characterization of α -carboxysomes from two different species indicates a variation of the CsoS2 distribution (Ni et al., Nat Commun 2022): in *Halothiobacillus* α -carboxysomes, CsoS2 is primarily occupying the Rubiscos close to the shell, whereas in *Cyanobium* α -carboxysomes, CsoS2 appears to be uniformly distributed. These observations indicate that CsoS2 may reach different depths to the α -carboxysome interior.

5. CsoS2-C dictates α -carboxysome shell assembly and architecture:

Regarding the shells missing pentameres: what could cause this exclusion of the pentamers from the assembly? This is not necessarily something to address experimentally, but it would be interesting to discuss.

We agree with the reviewer that the missing pentamer in the CsoS2-C3/4A-1A construct is very intriguing. Such pentamer-lacking synthetic BMC shells has also been observed by Sutter et al., ACS synthetic Biology 2019. The shells missing pentamer appear only observed

with the S2-C3 construct. Therefore, without the other fragments of S2-C for assembly of larger mini-shells, the population is shifted to an assembly where facets of CsoS1A are yet to be capped by CsoS4A pentamers, producing the pentamer-less shells as assembly intermediates. As indicated by immunoblot analysis (Figure 4 and Figure S9), the shell assembly with this construct is least efficient or least stable. Perhaps the F3 fragment is not sufficient to hold the pentamers onto the shell, thus departing together with the pentamer. Additional CsoS2 fragments (F1 and F2) help to tether F3 and pentamers onto the shell.

6. Regarding the western blot of CsoS1A in the pellet and supernatant to assess the fraction of protein assembled in shells (Fig 4b and Fig. S9). What is the reason for substantially higher expression of CsoS1A in S2-C2/4A-1A (and S2-C2/4A-1A)? Judging from the western-blot all constructs form essentially the same amount of mini-shells, and the relative ratio of assembly seems to mostly correlate with CsoS1A expression. Does probing for the other components yield the same results, i.e. do expression levels of the S2-variants correlate with CsoS1A expression? or is S2 expressed in the same amount in all constructs and S2 concentration is what dictates the amount of shell in the pellet? In other words, to draw conclusions regarding assembly efficiency between the different constructs it is important to demonstrate that S2 and CsoS1A are expressed in the same ratio for all constructs. One would expect a similar result with all components.

Considering the varying expression levels between constructs which could affect the amount of soluble proteins for mini-shell assembly, we determined the ratios of the amount of pelleted CsoS1A (representing CsoS1A in assembled mini-shells) to unpelleted CsoS1A (representing free CsoS1A hexamers or subunits), as the values plotted in Fig. 4b, representing 'relative ratio of assemblies/free proteins'. Since we focused on such ratios, we did not normalise the total protein content loaded per SDS-PAGE well or the number of cells (optical density at 600 nm) harvested for each expressed construct. We acknowledge that the use of the term 'assembly efficiency' was inappropriate in this context and have replaced it with 'the ratio of assembled shell proteins to unassembled shell proteins' in the text.

In this Western immunoblot, we tried to normalise the amount of assembled mini-shells across the constructs to get a visual representation of the assembly ratios, which is why it would seem like the amount of assembled mini-shells are the same for all the constructs. The S2-C2/4A-1A and S2-C3/4A-1A constructs have more unassembled CsoS1A and hence have the lowest ratios (Fig. 4b). The words 'pellet' and 'supernatant' here refer to that from a 30% sucrose cushion ultracentrifugation (Line303-305), but not the pellet and supernatant following clarification after cell lysis. Hence the amount of CsoS1A in the supernatant is not indicative of expression levels of the construct.

7. Has His79 on S1B (and S4B) been mutated to see how this affects assembly? Except for the hydrogen bond between His79 and S2-Tyr, are there any other important types of molecular interactions that contribute to the binding? Further, are there any theories of why F3 specifically interacts with the pentamer-hexamer interface while F1 and F2 do not?

We did not mutate the His79 on CsoS1A. Mutation was done on the CsoS2 [IV]TG motif to demonstrate its importance in shell assembly. There are many other specific interactions in all these interfaces such as the salt bridge between CsoS2 R731 and CsoS4A E22 (Figure3c), CsoS2 R784 and CsoS4A E69 (Figure 3d) and hydrogen bond between CosS2 R855 and CsoS4A H21. We focused on the repeating CsoS2 [IV]TG motif interacting with the shell

proteins, where the hydrogen bond between His79 and S2-Thr is one of the common interactions contributing to all the interfaces. In addition to two pairs of hydrogen bonds are formed by the main chains C=O and N-H groups, making a pair of short parallel β -strand

The [IV]TG motif mediates the interaction with CsoS1A in the same region (the β -strand which contains CsoS4A His79). Other sequences in CsoS2 before and after the [IV]TG motif are responsible for the specific interaction. For example, the hydrogen bond between CsoS1A His21 and CsoS2 D855 is specific in this pentamer-hexamer interfaces. On the other hand, the interaction appears to be related to local environment as well. In the T=7 Q=6 and T=4 Q=6 shells, F3 fragment does not appear to bind to pentamer-hexamer interfaces (FigS10).

8. CsoS2 exists as two isoforms transcribed from the same gene: CsoS2B - the full-length protein and CsoS2A - containing only the middle region and the N-terminal region and not the C-terminal shell binding motifs. It is unclear to us if CsoS2B/A or only CsoS2B has been used. CsoS2B would likely be the best for comparing mini-shell 2 with S2-C1, S2-C2 and S2-C2 constructs. If expressing both CsoS2B and 2A in mini-shell 2, what was the rationale for this? And how could this affect assembly? (By introducing a mutation removing the frameshift in the WT sequence one can produce the CsoS2B form. Chaijarasphong et al., 2016)

We used the wild-type full-length *csoS2* gene including its natural ribosomal frameshifting region to generate the minishell constructs. Therefore, both CsoS2A and CsoS2B isoforms can be expressed using the constructs.

To address the reviewer's question, we have performed immunoblot analysis using home-made anti-CsoS2 antibody (based on N-terminal Arg187- Gly200 fragment of CsoS2). As shown below, the results revealed that both CsoS2A and CsoS2B were expressed in the *E. coli* mini-shell construct, and the ratio of CsoS2A and CsoS2B is comparable to that found in the native α -carboxysome from *Halothiobacillus* (Sun et al., mBio 2022); by contrast, only CsoS2B was incorporated into the isolated mini-shells, confirming the importance of the CsoS2 C-terminus in attaching to the shell. We observe that there is a certain level of degraded CsoS2B in the purified sample. This is likely due to a long purification procedure (3 days). We have added the results in Figure S2b and the details in Methods.

Figure R5 | Immunoblot analysis of the content of CsoS2B and CsoS2A in mini-shells using home-made anti-CsoS2-N antibody (1:10,000 dilution; synthesized by GenScript, USA with the RGTRAVPPKQSQG peptide). WC: whole cell lysate of the CsoS2-CsoS4A-CsoS1A

mini-shell construct (mini-shell 2 construct); S2: isolated CsoS2-CsoS4A-CsoS1A mini-shells.

9. Related to point 10, a discussion about the role of the middle region in the assembly would be interesting. As is pointed out, this region contains the same motif as the C-terminal - can the middle region also interact with the shell? AlphaFold prediction for CsoS2 shows many secondary structures (albeit with various levels of confidence). Was there any indication that some regions of CsoS2 other than the C-ter could be ordered in these shells? Do the 2D class averages of the mini-shell 2 T=9 show any density (even fuzzy) inside the shell? Or is the rest of CsoS2 completely disordered? We understand that experiments on this are out of the scope of the paper, this is more of a curiosity and short discussion would be nice to add.

The middle region of CsoS2 contains several [IV]TG motifs as well (Figure S8b), which indicates that this region might interact with shell as well. However, our 2D classification did not reveal ordered densities within the shell (Figure R2a), and further symmetry expansion and focused classification did not reveal other binding sites either. We included these results in the Methods (Line 365 and Line 376-377). The local geometry of capsomer interfaces in the $T = 9$ shell can be different to that from native carboxysomes. It is possible that CsoS2 prefers to bind to these curved capsomer interfaces but would retain a certain affinity to flat interfaces, which is the case in the native carboxysomes facets. The exact role of CsoS2 middle region remains to be determined. We have added a discussion in the revised manuscript (Line 240-243).

10. The author should carefully check their references to make sure there is a sound balance between self-citation and citing others' work. As it is now 12 out of 18 carboxysome references are self-citations. (counting refs. 1-30, as 31-44 are method refs.). Relevant papers from other groups are missing, and in some cases reviews are cited where an original reference would be more suitable.

We have updated the citations by adding relevant references and original publications in the revised manuscript.

Methods

11. CryoEM data collection

Please specify the glow discharge parameters: pressure, which gas was used (likely ambient air, but specify), current and duration.

Please fully specify the type of grids: were they gold or copper grids? What was the mesh size?

We have included details of the experiments: "...The grids (Quantifoil™ R 2/1 on 300 copper mesh) were glow-discharged using a plasma cleaner (Harrick Plasma) with the plasma level set to High position for 45 seconds using ambient air. Three microliters of mini-shell samples were applied to the grids and blotted with filter paper for 3 seconds..."

12. CryoEM data processing: Please specify the exact versions of all software programs used:

- RELION
- cryoSPARC
- MotionCor2
- CTFFIND4

The software versions are specified in Methods.

13, Line 353: specify the value of the sharpening B-factor used for post-processing.

The B-factor used for map sharpening is -47.66, automatically determined by `relion_postprocess`.

14. Model building and refinement: Please specify exact versions of all software programs used:

- Coot
- Phenix.refine (version of the phenix suite)
- Chimera
- ChimeraX

The software versions are now indicated in Methods.

15. EMDB/PDB validation reports: Please consider depositing half maps and masks used for refinement and FSC calculation for all 15 structures deposited to the EMDB. They are required to calculate the masked FSC curve and thus to independently verify the resolution claims. Half maps and masks are also important for software and methods developers, so it benefits the whole cryo-EM community when they are deposited. Unfortunately, the EMDB does not make this a strict requirement. Of the 15 EMDB validation reports provided, only two are marked as “for manuscript review” (EMD-15595 and EMD-15611), while the others are draft validation reports with many fields left empty. While this does not raise much doubt about the validity of the data, it also prevents a complete examination. Please submit the final EMDB validation reports.

The half maps have submitted to EMDB and the final EMDB validation reports are now included.

Finally, we encourage the authors to deposit their raw movies into the EMPIAR database, at least for the datasets that yielded the 3 structures with an atomic model deposited in the PDB. We do not consider this a strict requirement for publication, but wish to promote this practice because it is highly beneficial for the cryo-EM community at large.

We have planned to deposit raw movies into the EMPIAR database. It is currently on-going.

Minor comments

Line 27: “Its mechanism of action, however, is not known”. This statement is not entirely true. Modify so it is clear that the mechanism is partly known e.g. C-term interacts with shell, N-term with Rubisco and details of N-term-Rubisco interactions are known.

Rephrased to: “Its mechanism of action, however, is not fully understood.”

Line 30: “the intrinsically disordered CsoS2 C-terminus is well-structured” sounds like a contradictory statement. Maybe try to rephrase.

Rephrased to: “The CsoS2 C-terminus was predicted to be intrinsically disordered, however is well-structured in the shell assembly.”

Line 40: we suggest adding biochemistry to this list of methods.

This is added now: "...advances in bioinformatics, biochemistry, imaging, and cell physiology ..."

Line 41: add that BMCs are composed entirely of proteins.

Added: "...bacterial microcompartments (BMCs) which is composed entirely of proteins, to compartmentalize metabolism..."

Line 59: "shell first" could be misinterpreted to believe that cargo enters the BMC after the shell is formed. Consider rephrasing this.

Thanks for pointing this out. We have revised this as "... 'Partial shell first' or 'Concomitant shell-core' assembly pathways ..."

Line 83: specify which form of CsoS2 was used, see comment 10.

We used the native *csoS2* gene from *Halothiobacillus*, preserving its ribosomal frameshift region. The SDS-PAGE of the purified mini-shell confirms that the majority of CsoS2 is in the full-length CsoS2B form, whereas the content of incorporated CsoS2A in mini-shells was under the detection level in immunoblot analysis (Figure R5 and Figure S2).

Lines 92-93: "A medium sized shell (~23 nm, $T = 4$) are present" should be edited to either "Medium sized shells [...] are present" or "A medium sized shell [...] is present".

Changed to: "A medium sized shell (~23 nm, $T = 4$) is present..."

Line 101: "extra densities not accounted for CsoS1A and CsoS4A" should be edited to "extra densities not accounted for by CsoS1A and CsoS4A".

Done.

Line 136: "CosS2" is a typo, should be "CsoS2".

Done

Line 139: change "arrangement" to "sequence" or "short linear motifs (SLiMs)"

Changed to "Repetitive short linear motifs have been identified in CsoS2..."

Line 169: do we know that the assembled shells are stabilized by S2 binding? The manuscript shows S2 involvement in the assembly process but does not provide evidence of higher stability of assembled shells. Consider rephrasing this. Same comment applies to Fig 3 legend.

We do not have experimental evidence demonstrating the assembled shells are stabilized by CsoS2 binding and therefore rephrased the claim to "...reinforcing the connections between capsomeres thus potentially stabilizing the assembled α -carboxysome shell."

The legend of Figure 3 is changed to: “CsoS2 binds to the shell through multivalent interactions...”

Lines 179-180: the statement that “the molecular threading of CsoS2-C is likely a universal feature” is a bold claim. Unless the authors can present data to support the “universal” character of this claim, we suggest rephrasing as follows (or similar, at the discretion of the authors): “the molecular threading of CsoS2-C is likely a widespread feature”.

Rephrased to “the molecular threading of CsoS2-C is likely a widespread feature”

Line 283: were TOP10 cells used for expression or is this a typo?

We have indeed used TOP10 cells for mini-shell expression in this study. TOP10 cells, unlike the BL21(DE3) expression strain, do not contain the enzyme to catabolise arabinose, which is the inducer for our mini-shell construct expression (pBAD). This ensures a steady concentration of inducer in the medium for stable expression of mini-shells over the entire expression period. We find that TOP10 cells work nicely and stably to produce mini-shells.

Line 291: “cell pellets” should be “pelleted mini-shells” (or similar)

Changed to “the pelleted mini-shells”

Lines 300-301: it would be helpful for readers to specify at roughly which salt concentration the mini-shells typically elute.

We have now added the details in the Methods section. “Mini-shells were eluted with a 0-40% linear gradient of buffer B (TEMB plus 1 M NaCl) and were found to be typically eluted at 30-35% buffer B. Purified samples were stored....”

Line 319: Change “ Five” to 5

changed.

Line 328: Change “Three” to 3

Changed.

Line 332: “a Gatan K3 director electron camera”. There is a typo.

Changed to “a Gatan K3 direct electron camera”

Line 345: “An ab initial model”. Typo, this should read “an ab initio model”.

Changed to “An *ab initio* model”

Line 379: “Initial models from crystal structures of hexamer (PDB 2EWH) and pentamer (PDB 2RCF).” This sentence needs a verb.

Changed to: “Initial models were obtained from crystal structures of hexamer...”

Line 380: Define which are the large shells.

Changed to: “For the $T = 9$ shell, the CsoS2 was traced manually into the density map...”

Figure 1a: split CsoS2 at $\frac{2}{3}$ (instead of $\frac{1}{2}$) as this better represent the size of S2B vs S2A

Moved CsoS2 splitting bar to $\frac{2}{3}$.

Figure 1b: specify if S2B or S2B/A was used for mini-shells

As responded above, we used the wild-type full-length *csoS2* gene including its natural frameshifting region to generate the mini-shell constructs. Therefore, both CsoS2A and CsoS2B isoforms were expressed using the constructs. However, unsurprisingly only CsoS2B was incorporated into the mini-shells. We have added the results in Figure S2b.

Figure S2a: due to other impurities on the gel can the band for CsoS2A be accurately assigned? If CsoS2A gets inside the carboxysome in mini-shell 2 this implies that other parts that the C-term of CsoS2 interacts with the shell or that CsoS2 self-interacts. If this is the case this should be commented on. See also comment 10.

We thank the reviewer for the comment and apologize for our oversight. Please see responses above. We only detected CsoS2B in the mini-shell sample, whereas CsoS2A was not detectable, confirming the importance of CsoS2 C-terminus in binding to the shell. We have added the results in Figure S2b.

Figure 2b, c: maybe use a different color scheme. Blue and orange are easy to distinguish, but gray is difficult to see.

We have changed grey to green for better presentation.

Figure 3c-e: add hydrogen bond between His79 and S2-Tyr. For easier visualization we suggest adding which fragment is shown in each panel instead of only in the figure legend.

Hydrogen bonds between His79 and S2-Thr are added and F1-F3 are labelled in the Figure3c-3e.

Figure 3f: add distance between His79 and S2-Tyr

A distance of 2.5 Å is labelled between His79 and S2-Thr.

Figure 4d: y-label is missing. what were the total numbers of shells that were counted to determine these proportions?

The y-label (“Percentage (%)”) is added. The quantification of shells is derived from cryo-EM analysis and the number of particles for such quantification is indicated in the Supplementary Table 1 and Supplementary Table 2 by “Final particle images (no.)” in each class. We added the following description in the figure legend: “Distribution of different shells in the four CsoS2 constructs as calculated from cryo-EM analysis; the number of particles for each shell is indicated in the Supplementary Table 1 and 2.”

Figure 4e: a small comment: it would be nice if this cartoon figure was somewhat in scale. Now S2 middle region and N-term are short in comparison to C-term and Rubisco is very big.

It is meant to be a cartoon illustration to show N- and C-terminus of CsoS2 interact with the Rubisco and the shell respectively. This off-scale allows to show both interactions clearly.

Figure S3a-b: Have these micrographs accidentally been switched?

Apologies for the mistake and now these two micrographs are correctly placed in the figure.

Figure S6: Which species are included in the clade named gammaproteobacteria? This is a class of bacteria while the other clades which are marked are grouped according to genus. Can this be a typo? Note that Halothiobacillus is a gammaproteobacteria.

Thanks for pointing this out. The highlighted gammaproteobacteria clade refers to several unclassified gammaproteobacterial bacteria. We have removed these from the figure and legend to avoid confusion. Figure S6 has been revised now.

Figure S7d: check that the RMSD values are correct. It looks as if 0.6 and 1.3 are swapped.

We apologize for the oversight. The RMSD values have been revised to 6.692, 14.377, and 4.232 in Figure S7c.

Figure S8c: how was the conservation score calculated? Using the ConSurf server? The conservation score was calculated using ConSurf server. We have included a description of this calculation in the figure legends: “The conservation score of CsoS2 fragments was calculated using ConSurf server (https://consurf.tau.ac.il/consurf_index.php)”.

Figure S10: maybe measure occupancy of CsoS2 in these maps with Occupy? See <https://doi.org/10.1101/2023.01.18.524529> and <https://occupy.readthedocs.io/en/latest> This would enable a quantitative assessment of whether CsoS2 is present or not, in addition to the visual assessment provided by the figure.

We appreciate the reviewer’s suggestion and have calculated the occupancy of the T=9 shell from the full-length CsoS2 and truncated CsoS2 constructs using Occupy. The rest of shells have relatively lower occupancy beyond reliable calculation using Occupy. We have included this measurement result as Figure S11.

Figure S8a: include distances of the hydrogen bonds. Which S2 fragment is shown here?

The distances of hydrogen bonds are now labelled. The first [IV]TG motif in F2 (I795-T796-G797) is shown in the figure. We add the description of this fragment in the figure legend: “A closeup view of one of the [IV]TG motif interfaces (I795-T796-G797 in F2) is shown on the right.”

Bioinformatics: it is good practice to include a SI table with all sequences used for building MSAs and phylogenetic trees.

The CsoS2 sequences for MSAs used in ConSurf server and phylogenetic tree are now included as supplementary files accompanying the manuscript (CsoS2_sequences_Consurf.txt and CsoS2 phylogenetic tree.fasta)

REVIEWERS' COMMENTS

Reviewer #1 (Remarks to the Author):

The authors answered all my points and changed the manuscript accordingly.

I am therefore favourable to the manuscript being published in its new form

Reviewer #2 (Remarks to the Author):

We thank the authors for their careful consideration of our review comments and are satisfied that these have been addressed. We recommend the paper for immediate publication.

Reviewer #3 (Remarks to the Author):

The authors have addressed my concerns and I support publishing the manuscript in Nature Communications.

Reviewer #4 (Remarks to the Author):

While the authors has addressed most of our concerns, a few remaining points are:

Major comments:

1. References: Please add Metskas, L.A., Ortega, D., Oltrogge, L.M. et al. Rubisco forms a lattice inside alpha-carboxysomes. Nat Commun 13, 4863 (2022) to the reference list. This paper is tomography of alpha-carboxysome as was published at the same time as the authors paper on the same topic (Ni et al, 2022) and they SHOULD be cited along with each other (Ni et al., 25th July, 2022 and Metskas et al., 18 Aug 2022 both in Nature Com).
2. Figure S9: Related to comment #6 in the initial review. It still says assembly efficiency in the figure legend in S9, update this to reflect the changes made in the manuscript. Please also add the explanation

of how this experiment was normalized to the figure legend (as is nicely described in the response letter).

3. Relating to comment #7 in review #1. Even if focus is on the [IV]TG motif the paper would greatly benefit from a discussion about other interactions involved in the binding. This does not have to be addressed experimentally but a discussion similar to what is written in the response letter should be added.

4. Bioinformatics:

Thank you for including the CsoS2 sequences. Please also include the sequences used for creating the MSAs for CsoS1A and CsoS4A.

Note that from a simple blast it will not be possible to distinguish CsoS1A from CsoS1C (differ in 2 aa from S1A) and CsoS1B (~95% identical to CsoS1A). The same thing goes for CsoS4A and CsoS4B. This should be noted in the figure legend. It will in ways also strengthen the results since it shows that the binding site is conserved among all 3 hexameric shell proteins and not in the 2 pentamers.

How big is the redundancy between the ~900 shell proteins? Sequences from databases are often biased towards species which have been heavily studied (eg. prochlorococcus). It can therefore be good practice to remove between 95-98% redundant sequences from such datasets (can easily be done in for example JalView). Has this been done and does that affect the results?

5. All validation reports are now the final ones (marked "for manuscript review"). But the masks used to calculate FSC curves have not been deposited (they are not listed in the validation reports, and they would be if they had been deposited). We insist that these masks should be deposited, at least for the three entries that have an associated atomic model (EMD-15798 and PDB 8B0Y; EMD-15799 and PDB 8B11; EMD-15801 and PDB 8B12), ideally for all 15 depositions.

Minor comments:

6. Line 30: From the previous review and response: "the intrinsically disordered CsoS2 C-terminus is well-structured" sounds like a contradictory statement. Maybe try to rephrase.

Rephrased to: "The CsoS2 C-terminus was predicted to be intrinsically disordered, however is well-structured in the shell assembly." This had not been changed in the submitted revision.

7. Figure 1b: Include the splitting bar on CsoS2 in mini-shell 2 (in other words, revise the same way as has been done in Figure 1a).

8. Figure S8c: Add the Consurf method to the method section as well.

9. Figure S8a: Add an annotation that blue is S1A and red S2.

10. A version number is still missing for the Phenix suite from which they used the phenix.refine program.

11. For RELION, the 2012 reference that is listed is fine but old. It would be better to list the ones for the version 3.1 that was used: <https://doi.org/10.1107/S2052252520000081> and <https://doi.org/10.7554/eLife.42166>

12. The Occupy program which used to add the new figure S11 is not cited: <https://doi.org/10.1101/2023.01.18.524529>

Reviewer #1 (Remarks to the Author):

The authors answered all my points and changed the manuscript accordingly. I am therefore favourable to the manuscript being published in its new form.

We thank the reviewer for their positive comments.

Reviewer #2 (Remarks to the Author):

We thank the authors for their careful consideration of our review comments and are satisfied that these have been addressed. We recommend the paper for immediate publication.

We thank the reviewer for their positive comments.

Reviewer #3 (Remarks to the Author):

The authors have addressed my concerns and I support publishing the manuscript in Nature Communications.

We thank the reviewer for their positive comments.

Reviewer #4 (Remarks to the Author):

While the authors has addressed most of our concerns, a few remaining points are:

Major comments:

1. References: Please add Metskas, L.A., Ortega, D., Oltrogge, L.M. et al. Rubisco forms a lattice inside alpha-carboxysomes. Nat Commun 13, 4863 (2022) to the reference list. This paper is tomography of alpha-carboxysome as was published at the same time as the authors paper on the same topic (Ni et al, 2022) and they SHOULD be cited along with each other (Ni et al., 25th July, 2022 and Metskas et al., 18 Aug 2022 both in Nature Com).

Reference has been now added.

2. Figure S9: Related to comment #6 in the initial review. It still says assembly efficiency in the figure legend in S9, update this to reflect the changes made in the manuscript. Please also add the explanation of how this experiment was normalized to the figure legend (as is nicely described in the response letter).

We have now revised the figure legend as follows:

“FigureS9|Western blot of minishell constructs with different CsoS2 mutants. The assembled shell (in the Pellet after 30% sucrose cushion ultracentrifugation) and free shell proteins (in the Supernatant after 30% sucrose cushion ultracentrifugation) were probed with anti-CsoS1A antibody. The ratios of assembled shell and free shell proteins are compared among the constructs (see Fig.4b). The immunoblot was normalised against the amount of assembled mini-shells across the constructs.”

3. Relating to comment #7 in review #1. Even if focus is on the [IV]TG motif the paper would greatly benefit from a discussion about other interactions involved in the binding. This does not have to be addressed experimentally but a discussion similar to what is written in the response letter should be added.

We have added a discussion about other interactions involved in the binding, in addition to the [IV]TG motif.

4. Bioinformatics:

Please also include the sequences used for creating the MSAs for CsoS1A and CsoS4A.

The files for CsoS1A and CsoS4A sequences have been provided.

Note that from a simple blast it will not be possible to distinguish CsoS1A from CsoS1C (differ in 2 aa from S1A) and CsoS1B (~91% identical to CsoS1A). The same thing goes for CsoS4A and CsoS4B. This should be noted in the figure legend. It will in ways also strengthen the results since it shows that the binding site is conserved among all 3 hexameric shell proteins and not in the 2 pentamers.

We agree with the reviewer that CsoS1A and CsoS1C have only two residues that are distinct from each other, and CsoS1A and CsoS1B are 91% identical. Only 40% similarity was found between CsoS4A and CsoS4B. Thus, we could not easily distinguish CsoS1A//B/C based on their sequences. We have revised Fig. S5a to CsoS1A/B/C but kept CsoS4A in Fig. S5b.

How big is the redundancy between the ~900 shell proteins? Sequences from databases are often biased towards species which have been heavily studied (eg. prochlorococcus). It can therefore be good practice to remove between 95-98% redundant sequences from such datasets (can easily be done in for example JalView). Has this been done and does that affect the results?

We selected the sequences carefully from the Uniprot databased and checked with JalView to remove any redundant sequences. The removal of the redundancy has little effect on the results.

5. All validation reports are now the final ones (marked "for manuscript review"). But the masks used to calculate FSC curves have not been deposited (they are not listed in the validation reports, and they would be if they had been deposited). We insist that these masks should be deposited, at least for the three entries that have an associated atomic model (EMD-15798 and PDB 8B0Y; EMD-15799 and PDB 8B11; EMD-15801 and PDB 8B12), ideally for all 15 depositions.

The masks for the three entries that have an associated atomic model are being deposited.

Minor comments:

6. Line 30: From the previous review and response: "the intrinsically disordered CsoS2 C-terminus is well-structured" sounds like a contradictory statement. Maybe try to rephrase.

Rephrased to: “The CsoS2 C-terminus was predicted to be intrinsically disordered, however is well-structured in the shell assembly.” This had not been changed in the submitted revision.

We appreciate the reviewer’s comment and have rephrased it accordingly.

7. Figure 1b: Include the splitting bar on CsoS2 in mini-shell 2 (in other words, revise the same way as has been done in Figure 1a).

Revised Figure 1b CsoS2 as suggested.

8. Figure S8c: Add the ConSurf method to the method section as well.

We added the following description of ConSurf to method section: “The conservation score of CsoS2 fragments is calculated using ConSurf server (https://consurf.tau.ac.il/consurf_index.php).”

9. Figure S8a: Add an annotation that blue is S1A and red S2.

We have added annotations that blue is S1A and red S2.

10. A version number is still missing for the Phenix suite from which they used the phenix.refine program.

Phenix version is now added (v1.19.2-4158) in the methods section.

11. For RELION, the 2012 reference that is listed is fine but old. It would be better to list the ones for the version 3.1 that was used: <https://doi.org/10.1107/S2052252520000081> and <https://doi.org/10.7554/eLife.42166>

The reference for RELION v3.1 is updated (Zivanov et al, Elife, 2018).

12. The Occupy program which used to add the new figure S11 is not cited: <https://doi.org/10.1101/2023.01.18.524529>

The Occupy program is now cited in the Methods section.